# UNCONDITIONAL CNN DENOISERS CONTAIN SPARSE SEMANTIC REPRESENTATIONS OF IMAGES

## ABSTRACT

Generative diffusion models learn probability densities over diverse image datasets by estimating the score with a neural network trained to remove noise. Despite their remarkable success in generating high-quality images, the internal mechanisms of the underlying score networks are not well understood. Here, we aim to understand the image representation that arises from score estimation objective. Unlike the vast majority of prior work, we focus on an *unconditionally* trained, fully-convolutional *UNet*, to isolate the role of the objective and architecture. We show that the middle block of the UNet decomposes individual images into sparse subsets of active channels, and that the vector of spatial averages of these channels can provide a nonlinear representation of the underlying clean images. Euclidean distances in this representation space are semantically meaningful, even though no conditioning information is provided during training. We develop a novel algorithm for stochastic reconstruction of images conditioned on this representation: The synthesis using the unconditional model is "self-guided" by the representation extracted from that very same model. For a given representation, the common patterns in the set of reconstructed samples reveal the features captured in the middle block of the UNet. Together, these results show, for the first time, that a measure of semantic similarity emerges, *unsupervised*, solely from the denoising objective.

## 1 INTRODUCTION

Generative diffusion methods provide a powerful framework for sampling from probability densities learned from complex high-dimensional data such as images (Sohl-Dickstein et al., 2015; Song and Ermon, 2019; Ho et al., 2020). The key to their success lies in deep neural networks trained to estimate the score (the gradient of the log of the noisy image distribution), which is achieved by optimization on a denoising task. Training and sampling algorithms for these models have been extensively studied (Croitoru et al., 2023), but the internal mechanisms and properties that enable their spectacular image generation capabilities are not understood. Estimating the score is equivalent to computing a minimum mean-square denoising estimator. Since denoising generally relies on distinguishing signal from noise, these score networks must somehow learn to identify and isolate patterns and structures found in their training data, so as to preserve them while eliminating noise. Here, we open the "black box" of a score network to reveal this internal representation.[1]

To isolate the role of the *objective*, we examine a UNet (Ronneberger et al., 2015), trained *unconditionally* for image denoising. The vast majority of prior work on image representation in diffusion models have explored models trained conditionally with top-down information, such as text, labels, representation from another model, similar/augmented image pairs, etc (Fuest et al., 2024). These studies reveal interesting properties about conditional denoisers, mostly driven by improving downstream applications. However, these representations are not only due to the score estimation objective, but confounded by the supervision imposed by the conditioner during training. These are parallel to supervised and self-supervised representation learning literature. Representation in an unconditional models is parallel to unsupervised representation learning methods such as sparse or variational autoencoder, which is difficult but arguably the ultimate goal of representation learning.

---

[1] Note that here the word "representation" is exclusively used in its standard way to refer to internal activations of the network, not to be confused with another usage of the word in diffusion model literature that refers to the intermediate points in the sampling trajectory, to or one-shot denoising, in pixel space (Graikos et al., 2023).

Prior work on *unconditional* models (Brempong et al., 2022; Yang and Wang, 2023; Xiang et al., 2023; Baranchuk et al., 2021; Mukhopadhyay et al., 2023; Tang et al., 2023) has shown that activations from such models can be extracted to perform downstream tasks – such as classification and segmentation – with considerable success. These results provide strong proof-of-concept that a semantic representation does exits in unconditional models. Since these works are driven by maximizing the utility of the representation for down-stream task, they all involve extensive search on hyper-parameters like optimal noise level, network block and pooling size for an intended task tuned to a given dataset. While this practice results in competitive performance, it does not go in the direction of interpretability. Our goal here is not to improve state-of-the-art on these downstream applications, but instead *to understand and interpret* those aspects of the internal representations of a denoiser at a mechanistic and geometric level.

We demonstrate that the unconditional UNet computes a low dimensional *sparse* set of activations in the output channels of the middle network block. This may be summarized with a vector comprised of the spatial averages of these channels, which provides a representation of the underlying clean image. We verify that this representation is stable when computed on an image contaminated with noise over a broad range of amplitudes.

This representation exhibits a number of intriguing properties. Roughly, the channels summarized by the representation vector fall into two categories: non-selective channels, which capture common features present in many images; and selective channels, which are specialized for patterns that occur in only a small subset of images. As a consequence, for diverse datasets, the representation vectors lie within a union of low-dimensional subspaces, each of which is spanned by many of the common channels and a small subset of the specialized channels. We demonstrate that distances in this space are meaningful: Images whose representation vectors are similar are also semantically similar. This allows us to partition the images in the dataset by applying a clustering algorithm to their representation vectors. The emergent clusters capture the general visual appearance of the corresponding images, sharing both fine details as well as global structure, but are only partially aligned with object category labels.

Finally, we develop an algorithm for stochastic sampling, using a reverse diffusion algorithm conditioned on this representation vector computed from a target image. This procedure recovers a sample from a set of images whose representation is the same as that of the target image. Visually, these images are similar in terms of both local and global patterns, revealing both the commonality and diversity of attributes encoded in the representation. To quantify this, we show that the Euclidean distance between a pair of representation vectors strongly predicts the distance between a pair of conditional distributions induced by the representations. Thus, we show that the denoising objective alone, without any external conditioning, engenders learning of high level features that carry detailed semantic information.

## 2 IMPLICIT SPARSE IMAGE REPRESENTATION

Diffusion models learn image densities from data using a network that is trained for denoising. Specifically, given a noisy image $x_\sigma = x + \sigma z$, with $z \in \mathcal{N}(0, \text{Id})$ a sample of white Gaussian noise, one trains a network $s_\theta(x_\sigma)$ to minimize the squared error:

$$\ell(\theta) = \mathbb{E}_{x,\sigma,z}\|x - \hat{x}(x_\sigma)\|^2 = \mathbb{E}_{x,\sigma,z}\|s_\theta(x_\sigma) - \sigma z\|^2. \tag{1}$$

The optimal solution is the conditional mean, which can be directly related to the score of the underlying noisy distributions using a relationship published in Miyasawa (1961), but generally referred to as "Tweedie's formula" (Robbins, 1956; Efron, 2011):

$$\hat{x}(x_\sigma) = \mathbb{E}[x|x_\sigma] = x_\sigma + \sigma^2 \nabla_{x_\sigma} \log p_\sigma(x_\sigma). \tag{2}$$

Thus, the trained network provides an approximation of the family of score functions for all $\sigma$. Reverse diffusion methods draw samples through iterative partial application of the learned denoiser, thereby using this approximate score to ascend the probability landscape (Sohl-Dickstein et al., 2015; Song et al., 2020; Ho et al., 2020; Kadkhodaie and Simoncelli, 2020).

### 2.1 SPATIAL AVERAGES OF ACTIVATIONS

We adopt a convolutional UNet architecture (Ronneberger et al., 2015), a smaller and more readily analyzed architecture than the most recent implementations, that nevertheless offers strong denoising

performance. The model consists of three main components: a set of encoder blocks (the downsampling path), a middle block, and a set of decoder blocks (the upsampling path) (see Appendix A for details). We examine the activations of these hidden layers to understand how the noisy image is transformed to produce the score. To reduce the dimensionality, we consider the simplest summary of channel activations, $a_j(x_\sigma)$, consisting of the spatial averages (i.e. global pooling) of all channels at the output layer of each block, notated by $\bar{a}_j(x_\sigma) \in \mathbb{R}^{d_j}$. The components of the vector $\bar{a}_j$ are nonnegative (due to half-wave rectification (ReLU) nonlinearities), and carry information about the features encoded by their corresponding channels. Because of the convolutional nature of the network, the activations within each channel indicate the presence or absence of a feature in different spatial locations. Thus if a feature associated with channel $i \in (0, d_j)$ is present somewhere in the input image, $\bar{a}_j(x_\sigma)[i]$ will be positive, otherwise zero.

It is worth noting that the structure of the UNet architecture imposes certain properties on $\bar{a}_j$. Idealized convolutions are translation equivariant, hence the spatial averages of their activations are translation *invariant* w.r.t to the extracted features and do not carry information about location. For ergodic processes, these averages are approximations of statistical moments (expected value of nonlinear functions), and such measurements have been used for representation of visual texture Julesz (1962); Zhu et al. (1998); Portilla and Simoncelli (2000); Victor et al. (2017). However, translation equivariance is imperfect for UNets (and most other convolutional networks) since it is violated by zero-padded boundary handling, and downsampling operations. Both effects are more prominent in the deeper blocks of the encoder, which have undergone more downsampling, and for which the boundary influence encroaches on a larger spatial portion of the channels. In these cases, $\bar{a}_j$ can carry information about the location of features in addition to their presence. This effect coincides with the growth of the receptive field (RF) in the deeper blocks. As a result, $\bar{a}_j$ is expected to carry location information for larger features (Kadkhodaie et al., 2023; Kamb and Ganguli, 2024).

## 2.2 Denoising and channel sparsity

One of the difficulties in studying image representation in diffusion models is that it is not obvious where to look. We aim to locate the layers whose feature vectors, $\bar{a}$'s, well represent the clean image underlying the noisy input image. Such feature vectors would be rich enough to capture patterns and regularities of the image behind distortions caused by the noise in the input. Unlike Variational AutoEncoders (VAEs) (Kingma et al., 2013), for which the encoder, bottleneck and decoder are defined by their distinct assigned roles through a dual objective function, the components of the UNet denoiser are only defined architecturally, and the optimization is solely driven by a single end-to-end denoising objective. As a result it is not clear where the denoising occurs.

Some insight arises from considering the operation of denoisers designed in the pre-DNN era. Traditionally, image denoisers operate by transforming the noisy image to a latent space in which the true image is concentrated (sparse), and the noise is distributed (dense). Then the noise is suppressed and the image preserved. Finally, the latent representation is transformed back to the image space. This basic description captures the Wiener filter (Wiener, 1964)(operating in the frequency domain), and thresholding denoisers which are typically applied within a multi-scale wavelet decomposition (Milanfar, 2012). In all cases, more concentrated signal representations lead to better separation of noise and signal, hence superior denoising.

In the era of deep learning, DNNs optimized for denoising far outperform traditional solutions. Assuming the same principles apply, we expect to observe an increase in representation sparsity in portions of the network responsible for removing noise. To search for this locus, we measure the sparsity of $\bar{a}$ at the input and output layers of each block of a UNet trained on ImageNet. Sparsity of $\bar{a}$ is quantified by the normalized *participation ratio* (PR), the squared ratio of $L_1$ and $L_2$ norms:

$$\text{PR}(\bar{a}) = \frac{\|\bar{a}\|_1^2}{d \, \|\bar{a}\|_2^2}.$$

where $d$ is the ambient dimensionality of the space, i.e. number of channels in the layer. $\text{PR} \in (0, 1]$, and provides a soft measure of dimensionality: small values reflect low dimensionality (sparsity), and large values indicate high dimensionality (density).

Figure 1 shows distributions of sparsity for input/output layers of blocks in the UNet denoiser trained on ImageNet64 dataset. Both encoding blocks offer a substantial decrease in sparsity of $\bar{a}$, but we

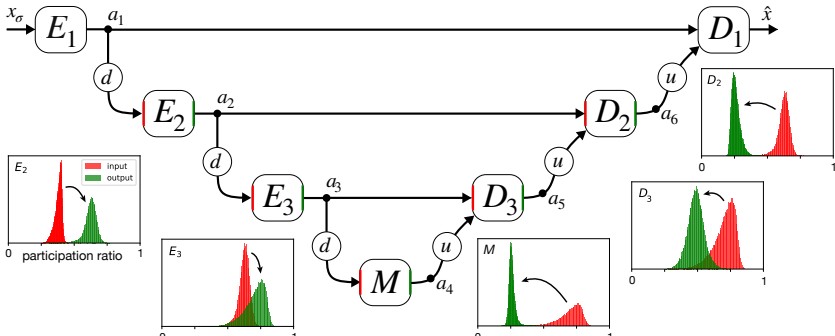

Figure 1: Channel sparsity of input and output layers of a UNet trained on ImageNet. Histograms show participation ratios (PR), of the spatially averaged input channels (orange) and output channels (green) for individual blocks. The middle block and decoder blocks exhibit increases in sparsity (i.e. reduction in PR). Blocks $\{E_1, D_1\}$ are not included since they have only one input/output channel, receptively. This is evidence that encoder blocks extract features to isolate noise and signal, and middle block and decoder blocks preserve those channels containing signal while suppressing those containing noise. (Notation: encoder blocks $\{E_k\}$, middle block $(M)$, decoder blocks $\{D_k\}$, downsampling $(d)$, upsampling $(u)$, "skip" connections.) See Figure 10 for other models.

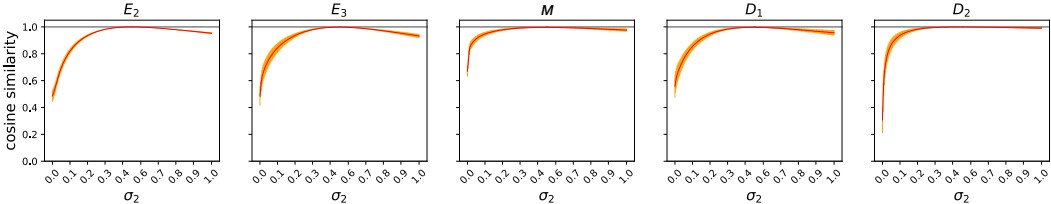

Figure 2: Stability of $\bar{a}$ across noise levels, for different network blocks of a model trained on ImageNet64. Plots show cosine similarity of $\bar{a}(x_{\sigma_1})$ and $\bar{a}(x_{\sigma_2})$, for $\sigma_1 = 0.5$, as a function of $\sigma_2$. $\bar{a}$ is most stable in the middle block (M). Note that $\bar{a}$ collapses as $\sigma$ falls to zero, for which the denoiser should compute the identity function. See Figure 11 for other models.

observe a stark increase of sparsity in the middle and decoding blocks. This suggests that removal of noise, through suppression of channels whose activations are primarily carrying noise, starts at the middle blocks, and then continues in the decoder blocks. The decoder block $j$, hence, denoises the features present in $a_j$ passed from encoder $j$, conditioned on the denoised features coming from below. This is evidence that the encoder transforms the noisy image to a hierarchical representation in preparation for denoising, by extracting image structures learned from the training set. Thus, the representation lends itself to investigation at the output layers of the middle and decoder blocks, where image features are exposed after removal of noise. Figure 10 shows this phenomenon in three additional UNet models trained on Texture, CelebA, and LSUN-Bedroom datasets.

## 2.3 ROBUSTNESS OF REPRESENTATION TO NOISE

To elucidate the relationship between $x$ and $\bar{a}$, we need first to clarify the effects of noise. The vector $\bar{a}$ depends on both the noise amplitude, $\sigma$, and the particular noise realization $x_\sigma$. In the context of diffusion sampling algorithms, the former translates to the evolution of representation with time. Not surprisingly, the variance in $\bar{a}$ grows with noise level. To remove this variability, we take the mean of $\bar{a}$ across noise realizations, $\mathbb{E}_z[\bar{a}(x + \sigma z)]$. Figures 2 and 11 show the effect of noise level on $\bar{a}$. Variability due to noise level depends on the block depth, but interestingly, $\bar{a}_j$ is most stable in the middle block: increased noise level mostly increases the amplitude but not the direction of $\bar{a}_j$. The noise resilience of $\bar{a}$ at the output layer of the middle block makes it a good candidate for study. Thus, for the remainder of this paper we focus on the representation in this layer, which we notate as:

$$\phi(x_\sigma) = \mathbb{E}_z[\bar{a}_4(x + \sigma z)].$$

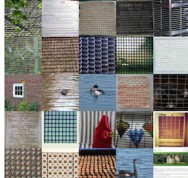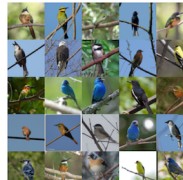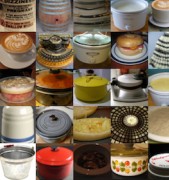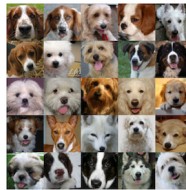

Figure 3: Channel selectivity. **Left:** Participation ratios for each channel over ImageNet. Distribution is bimodal, corresponding to channels that are highly specialized (and infrequently active) on left, and commonly used on right. **Right:** The panels show the set of images that maximally activate each of four specialized channels, revealing selectivity for rectangular periodic lattices (PR$= 0.19$), a bird on a branch (0.18), cylindrical objects(0.19), and dog faces (0.26). See Figure 12 for other models.

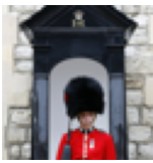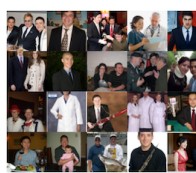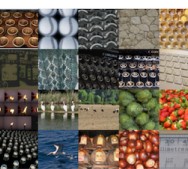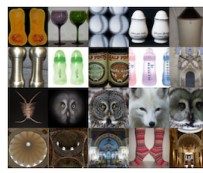

Figure 4: Specialized channels capture visual attributes and composition of an image. **Left:** Example image that activates several specialized channels. **Right:** Each panel shows the set of images that maximally activate one of the specialized channels activated by the example image, corresponding to images of people, periodic texture patterns, and images with left-right reflective symmetry. All three elements are present in the example image.

## 2.4 Denoising and channel selectivity

In this section, we examine the components of the representation vector $\phi$, and their relationship to the content of the (clean) image $x$. For each channel, $i$, in the output layer of the middle block we quantify its (non-)selectivity using the participation ratio: $\frac{\|c(i)\|_1^2}{n\,\|c(i)\|_2^2}$, where $c(i) = (\phi(x_{1\sigma})[i], ..., \phi(x_{n\sigma})[i])$ is the concatenation of the $i^{th}$ entry of $\phi$ for all $n$ images in the dataset. This provides an estimate of the number of active channels, with smaller values indicating more selectivity. Figure 3 shows the distribution of selectivity for all 512 channels in the middle block of a UNet trained on ImageNet64. The distribution is bimodal. Channels fall roughly into one of the two categories: *selective channels* that respond only to a small fraction of images in the set, and *non-selective channels* that respond to many images.

Specialized channels respond to specific features or patterns, and have negligible responses for most images in the dataset. These are akin to object detector neurons found in scene classifiers and GANs (Bau et al., 2020). Figure 3 shows sets of images that maximally excite four example specialized channels. The pattern shared across these images indicates the feature extracted by that channel. Since each image only contains a subset of patterns out of all the patterns present in the data distribution, only a handful of specialized channels are activated for each image. Figure 4 shows an example image along with the top three specialized channels it activates. Each channel is selective for a particular property of the image. Over the entire dataset, all channels are used, but different channels are used for different images.

More generally, channel selectivity predicts some statistical properties of the channels (Figure 13). The marginal distribution of $\phi$ values in specialized channels is heavy-tailed, but common channels are closer to Gaussian. Specialized channels are spatially sparse while common channels are spatially denser. Additionally, PCA analysis on activation maps of specialized channels shows that they are highly concentrated in a few directions while the common channels are explained with more dimensions. Finally, less selective channels respond to a larger set of frequencies and orientations, and are the most stable w.r.t to noise level. These results imply that the common channels capture shared or common image features, such as brightness, global structure of a scene, etc.

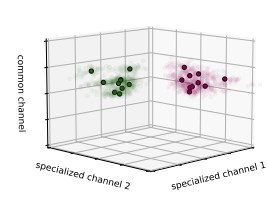 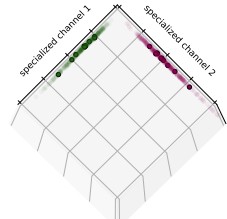

Figure 5: Union of subspaces. **Left:** Two sets of images whose $\phi$'s lie on two subspaces. **Middle/Right:** Three components of the $\phi$ vectors (out of the 512) for these images. The vertical axis corresponds to a common channel, while the other two correspond to specialized channels, each selective for only one image cluster. As a result, the $\phi$ vectors lie on a union of two-dimensional subspaces in the displayed three dimensional ambient space.

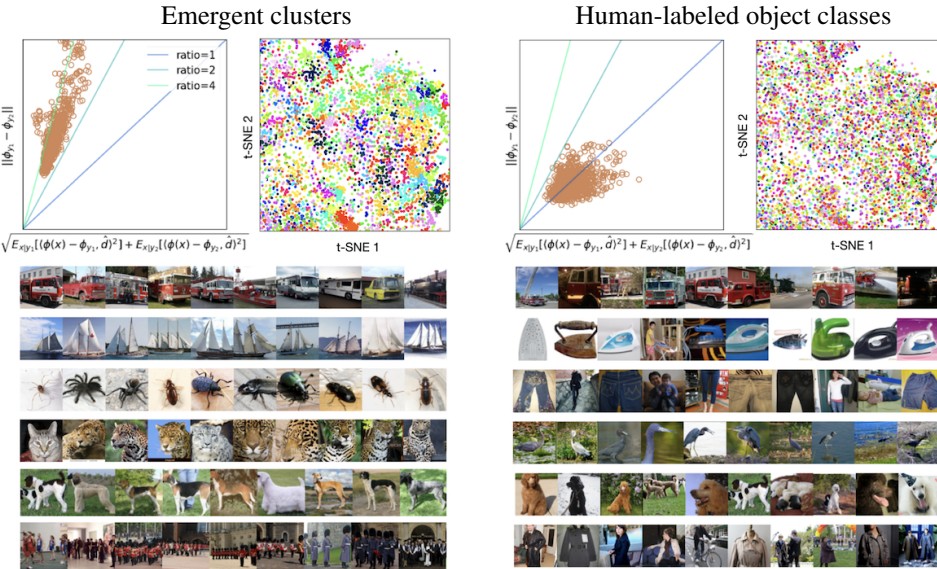

Figure 6: Clusters of representation vectors, $\phi$, are strongly concentrated. Comparison with class labels reveals that clusters capture semantic abstractions which are only partially aligned by object identity. **Left panel:** Upper left scatterplot shows separation of random pairs of clusters using d-prime. For all pairs, centroids are more than two standard deviations apart, and thus well-separated. Upper right is a t-SNE (Maaten and Hinton, 2008) visualization of a subset of the space. Each dot is a two dimensional projection of $\phi$, color-coded by cluster. Each row contains images randomly selected from a cluster. Images within a cluster are visually similar and share global organization and semantic patterns, but they not necessarily from the same class. **Right panel:** Analogous plots for pairs of human-labeled classes, which are substantially more overlapping in the representation space. Random examples of images within 6 classes are shown for comparison. More cluster examples in Appendix B.

## 2.5 DENOISING AND EMERGENT SEMANTIC SIMILARITY

An immediate consequence of *sparsity* and *channel selectivity* is that the $\phi$'s lie in a *union of subspaces* in $\mathbb{R}^d$, whose dimensionality lies approximately in the range $[0.2d, 0.3d]$ (from Figure 1). Each subspace corresponds to a combination of features that are likely to occur simultaneously (Figure 5). What do images whose $\phi$'s lie on the same subspace have in common? To examine this, we gathered images from the dataset with highest cosine similarity to a given target image in the representation space. Figures 5 and Figure 14 show two sets of those images. Images that are closest in the $\phi$ are semantically similar, in stark contrast to images that are closest in the pixel space.

The fact that pairwise Euclidean distances in the $\phi$ space are meaningful implies that similar images cluster together within the subspace. To test this, we applied a K-Means clustering algorithm (Lloyd, 1982), with $K = 1000$, to the $\phi$ vectors computed from the ImageNet dataset (See Appendix B). Upper left Scatter plot in Figure 6 shows that within clusters the $\phi$'s are well concentrated around centroids, and the centroids are well separated, measured by d-prime, a standard separability measure in Signal Detection Theory:

$$d' = \frac{\|\phi_1 - \phi_2\|}{\sqrt{\mathbb{E}[\langle \phi(x_\sigma) - \phi_1, \hat{d}\rangle^2] + \mathbb{E}[\langle \phi(x_\sigma) - \phi_2, \hat{d}\rangle^2]}}$$

where $\phi_1$ and $\phi_2$ denote the centroids of two clusters, and $\hat{d}$ denotes the line connecting them. The numerator measures the distance between the centroids of two clusters, and the denominator measures the average variance of the two clusters along the line that connects them. Larger ratio corresponds to more separated clusters. The scatterplot on the left shows that centroids of adjacent clusters are at least 2 standard deviations apart. The scatter plot on the right shows that the $\phi$'s of human labeled classes, by comparison, are less well separated. As a result, we expect images from different classes to appear within the same cluster.

Each row in the left panel of Figure 6 shows random images within a cluster. These images are semantically similar: they clearly share location-specific global scene structure, as well as location-nonspecific detailed elements (e.g., objects and their constituent parts, texture). However, in many cases, the similarity is not the object identity: a variety of trucks appear in one cluster, dogs on a grass background in another, and a variety of bugs on a flat background in another. This analysis explains why a linear classification of $\phi(x_\sigma)$ is significantly less accurate compared to non-linear classification (Mukhopadhyay et al., 2023). Importantly, this reveals that the semantic abstraction that is useful for computing the score are not object identities, hence the natural grouping that arises from training is not fully aligned with class-labels. From an information theoretic perspective, class labels do not have high mutual information with the image (Premkumar, 2025). Instead, to minimize the loss, the internal representation is structured such that images with  What connects images within the same cluster is "the gist of the scene" (Potter and Levy, 1969; Oliva and Torralba, 2001; Oliva, 2005; Oliva and Torralba, 2006; Sanocki et al., 2023), which is only partially consistent with human labeled class memberships. Remarkably, these *emergent clusters* arise solely from learning the score, by minimizing a denoising objective. The network is trained without labels, augmentations, or regularization to induce any specific type of similarity or grouping (e.g membership in the same class). Therefore proximity in the representation space defines a *fully unsupervised* form of similarity in the image space.

## 3    STOCHASTIC IMAGE RECONSTRUCTION BY CONDITIONAL SAMPLING

In order to define the representation more precisely, we propose an algorithm to "decode" $\phi$. As with any other representation, decoding $\phi$ is essential to exposing what it represents. Computing $\phi$ involves taking spatial average of the activations. Hence, $\phi$ represents not only the image it is computed from, but a whole set of images which have different activations with the same spatial average. In other words, $\phi$ is a many to one function. The goal is to sample from $p(x)$ using a diffusion algorithm, while requiring that all samples have the same $\phi(x^c)$ computed from a target image, $x^c$. Since the algorithm samples images that are consistent with the network's own representation of a target image, it is a *stochastic* reconstruction from the representation.

The algorithm is obtained by augmenting a reverse diffusion algorithm with a soft projection onto the non-convex set in pixel space defined by $\phi$. To achieve this, the score step is alternated with an iterative projection step, where the sample is modified until its representation matches $\phi(x^c)$. This matching step is analogous to methods used to generate texture images from their measured statistics (Portilla and Simoncelli, 2000). The matching is implemented by minimizing the Euclidean distance between the sample and target representations, $\|\phi(x) - \phi(x^c)\|^2$, which is achieved by back-propagating the gradient of the loss through the first half of the network (i.e. $\bar{a}_4(x_\sigma)$). The algorithm, hence, consists of two alternating steps described in Algorithms 2 and 1: At every time point in the synthesis, first the $\phi$ of the sample is matched to the target's, via back propagating a gradient in the score network. Then, the score-directed step pushes the sample closer to $p(x)$ by removing noise via using the decoder of the UNet.

---

**Algorithm 1** Stochastic reconstruction from the representation

---

**Require:** score network $s$, conditioner image $x^c$, sigma schedule $(\sigma_T, ..., \sigma_0)$, distribution mean $m$
    Draw $x_T \sim \mathcal{N}(m, \sigma_T^2 \mathrm{Id})$                                 $\triangleright$ Initialize sample
    **for** $t \in (T, ..., 1)$ **do**                 $\triangleright$ Alternate between matching and score steps
        Draw $z \sim \mathcal{N}(0, \mathrm{Id})$
        $x_t^c = x_c + \sigma_t z$
        $x_t \leftarrow$ Guidance step $(x_t, x_t^c)$         $\triangleright$ Update $x_t$ to match representations using Algorithm 2
        Draw $z \sim \mathcal{N}(0, I)$
        $x_t \leftarrow x_t + s(x_t) + \sigma_{t-1} z$                   $\triangleright$ Update $x_t$ using the score
    **end for**
    $x \leftarrow x_1 + s(x_1)$
    **return** $x$

---

**Algorithm 2** Guidance step

---

**Require:** score network $s$ to compute $\phi(x)$, sample $x_t$, conditioner image $x_t^c$, learning rate of optimizer $\eta$
    $\phi_{x_t^c} = \phi(x_t^c)$ and $\phi_{x_t} = \phi(x_t)$                     $\triangleright$ Compute representations
    $\mathcal{L}(x_t) = \|\phi(x_t) - \phi_{x_t^c}\|^2$               $\triangleright$ Compute distance between representations

    **while** not converged **do**           $\triangleright$ Minimize distance by backpropagating gradient through $e$
        $\nabla_{x_t} \mathcal{L}(x_t) = (\phi_{x_t^c} - \phi(x_t))^T \nabla_{x_t} \phi(x_t)$        $\triangleright$ Compute gradient
        $x_t \leftarrow x_t - \eta \nabla_{x_t} \mathcal{L}(x_t)$                     $\triangleright$ Update
    **end while**
    **return** $x_t$

---

This procedure can be described as a "self-guided" sampling algorithm , where the synthesis is guided by the network's own representation. Matching the $\phi$'s changes the trajectory by forcing the sample to be within the set of images whose $\phi$ vectors are equal to the conditioner's. The matching step does not change the noise level (Figure 20), so it is in a sense "orthogonal" to the score step. This means that matching iterations guide the sample in the domain of images that are high probability according to $p_\sigma(x_\sigma)$ to reach the set of images defined by $\phi(x_\sigma^c)$. A conceptual diagram of the algorithm is shown in Figure 8.

Figure 7 shows samples generated by the algorithm. In each panel, the representation is obtained from a real image. Then the stochastic reconstruction algorithm is used to generate 8 images with the same $\phi$. The conditionally generated samples are not identical, but all are visually similar to the conditioner image $x^c$. Importantly, they share location-specific global structure as well as location-nonspecific details. Thus, the visual commonalities and diversities in the samples reveal what is and is not captured by the spatial averages of the feature vector in the middle block.

Each $\phi$ characterizes a conditional density, $p(x|\phi)$, from which Algorithm 1 draws samples. We show that Euclidean distances between $\phi$'s are correlated with distances between conditional densities they induce. This is captured by a Euclidean embedding property, which ensures that the separation of $\phi$'s is related to a distance between the probability distributions $p(x|\phi)$, and hence that there exists $0 < A \le B$ with $B/A$ not too large, such that

$$\forall x_1, x_2 \quad , \quad A\|\phi(x_1) - \phi(x_2)\|^2 \le d^2(p_1, p_2) \le B\|\phi(x_1) - \phi(x_2)\|^2. \tag{3}$$

We establish a distance between two conditional densities as

$$d^2(p_1, p_2) = \int_0^\infty \left( \mathbb{E}_{p_1}\left[\|\nabla \log p_1 - \nabla \log p_2\|^2\right] + \mathbb{E}_{p_2}\left[\|\nabla \log p_1 - \nabla \log p_2\|^2\right] \right) \sigma \, d\sigma. \tag{4}$$

where $p_1 = p_\sigma(x_\sigma|\phi_1)$ and $p_2 = p_\sigma(x_\sigma|\phi_2)$. This distance is based on the difference in the expected score assigned to $x_\sigma$ by $p(x_\sigma|\phi_1)$ versus $p(x_\sigma|\phi_2)$, integrated across all noise levels. It provide a distance by symmetrizing the Kullback-Leibler divergence $\mathrm{KL}(p\|q)$ between two distributions $p$ and $q$ proved in (Verdú, 2010; Lyu, 2012):

$$\mathrm{KL}(p\|q) = \int_0^\infty \mathbb{E}_{p_\sigma}[\|\nabla_{x_\sigma} \log p_\sigma(x_\sigma) - \nabla_{x_\sigma} \log q_\sigma(x_\sigma)\|^2]\,\sigma d\sigma.$$

The right panel of Figure 8 confirms that, for a set of random pairs of images, Euclidean distances of their representations satisfy the Euclidean embedding inequality (3) with $\frac{B}{A} = 2.9$. See fig. 21 for an intuition about what the symmetrized distance measures.

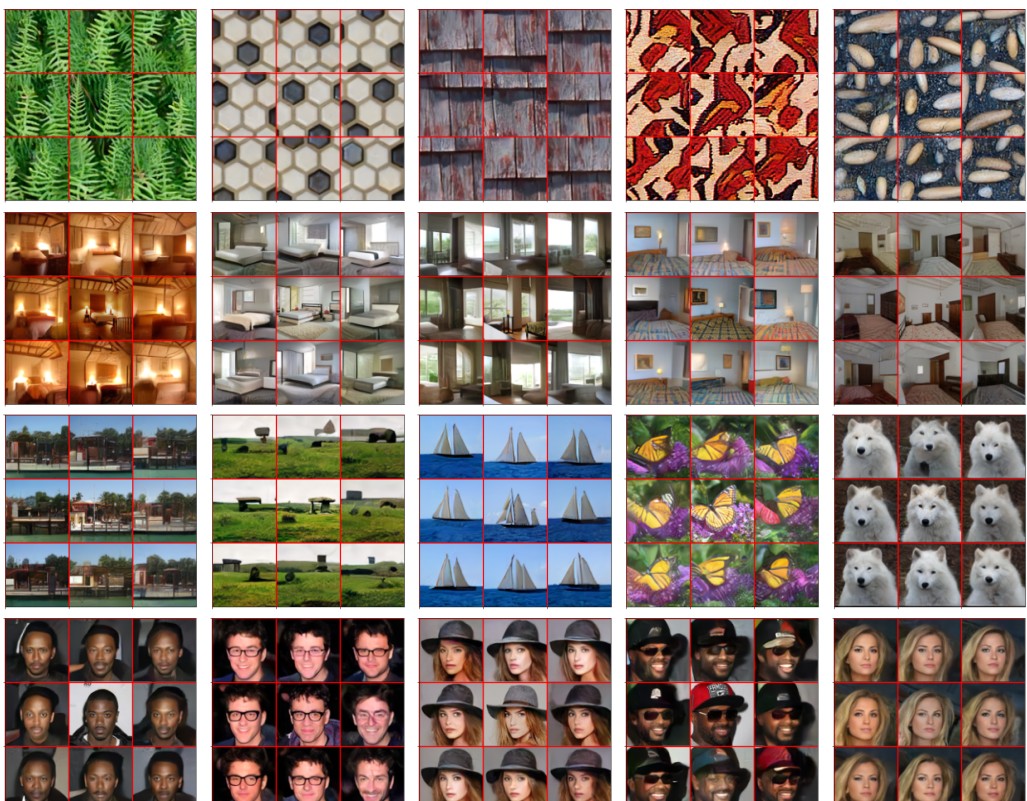

Figure 7: Samples from the stochastic reconstruction algorithm 1, using models trained *unconditionally* on Texture, LSUN-Bedrooms, ImageNet64, and CelebA datasets (from top to bottom row). Image at center of each panel is an original image from which the target $\phi$ is computed. The surrounding eight images are samples, conditioned on that $\phi$. Semantic similarities between samples within each panel reveal image structures captured by $\phi$. See Appendix C for more examples and also samples from the same models without conditioning.

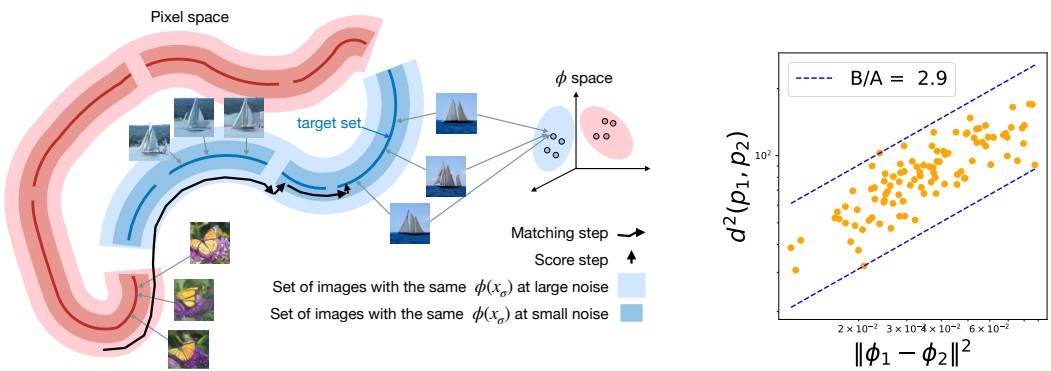

Figure 8: **Left:** Schematic of alternating matching and score steps. The support of $p(x)$ consists of eight manifolds, each containing a set of images with the same $\phi$. Same color $\phi$'s lie on the same subspace in the representation space. The shaded areas show support of $p(x_\sigma)$ at two noise levels: at very high noise, all the four blue sets have the same $\phi$ and at intermediate noise every two manifolds have the same $\phi$. The arrows show the sampling trajectory. The matching steps move along the manifold and do not change the noise level and come to a halt when entering the target $\phi$ set. The score step move towards the manifold by reducing the noise level. **Right** The distance between random pairs of $\phi$ vectors is strongly related to the distance between the pair of conditional densities they induce.

## 4 DISCUSSION

Generative diffusion models have shown incredible success in learning and sampling from image densities. This feat is due to near minimum mean square error of denoising networks used in these algorithms. We hypothesized that these networks must construct an internal representation of image features that differentiate signal from noise, and we sought to elucidate that representation. We found that the vector of spatial averages of channels in the middle block of a trained UNet can provide a signature of this representation. The components of this vector are sparse and thus the scores of complex image distributions have a low-dimensional structure, which is made explicit in these networks. We also found that the geometry of the representation space is meaningful and nearby images in representation space are semantically similar in pixel space. We showed that unsupervised clustering of the representation vectors yields well-separated groups of images that are semantically related, but only partially aligned with object identity. We developed a stochastic reconstruction algorithm, and showed that Euclidean distances in representation space are correlated with the symmetrized KL divergence of reconstruction distributions. These results show that a network trained "bottom-up", using only a denoising objective and no external conditioning, labeling, augmentation or regularization, can nevertheless learn a rich and accessible representation of image structure.

**Limitations and future directions.** Many open questions remain to be explored. One concerns a deeper understanding of the geometry of the representation space. We observed that the denoiser transforms a union of manifolds into a union of subspaces. What is the distribution of the dimensionalities of these subspaces, and how many of them, out of all possible ones, are spanned by the $\phi$'s? Importantly, can we estimate the joint distribution of $\phi$'s within these subspaces? A more comprehensive understanding of the latent space could be used to generalize Algorithm 1 to other conditional settings—for example, sampling from an emergent cluster given its centroid, or combining features to create new subspaces, thereby enabling out-of-distribution generalization. Another direction is to investigate the role of the encoder and decoder blocks in representing image structure, and how these representations depend on the noise level. This requires analyzing the interaction between layer depth and noise level, a relationship that is complex but essential to characterize. Finally, while our analysis has focused on fully convolutional neural networks, it is important to examine whether similar representations with comparable properties also arise in more modern architectures such as transformers.

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

# A  ARCHITECTURE, DATASET, TRAINING

## A.1  ARCHITECTURE

We used the original UNet architecture (Ronneberger et al., 2015) with a few modifications. The architecture is made up of 3 encoder blocks, 1 middle block, and 3 decoder blocks. The number of layers in each block is 2, 3, and 6 for the encoder, middle and decoder blocks. Each layer consists of $3 \times 3$ convolutions with zero boundary handling, Layer normalization and ReLU. The number of channels is 64 in the first encoder block and then grows by a factor of two after each downsampling operation, and decreased by a factor of two after each upsampling operation. This results in 64, 128, 256 channels in the encoder blocks, 512 channels in the middle block, and 256, 128, 64 in the decoding blocks. The total number of parameters is $\sim 13m$.

The **receptive field size** at the end of the middle block is $84 \times 84$ surpassing the spatial size of the input images in the ImagNet dataset. This is required for capturing large structures in the absence of attention blocks (Kadkhodaie et al., 2023). The depth of decoder blocks is increased in order to increase the expressivity of the denoising operations. This is again related to the size of the receptive field at the end of each decoder block with respect to the representation at the end of the middle block.

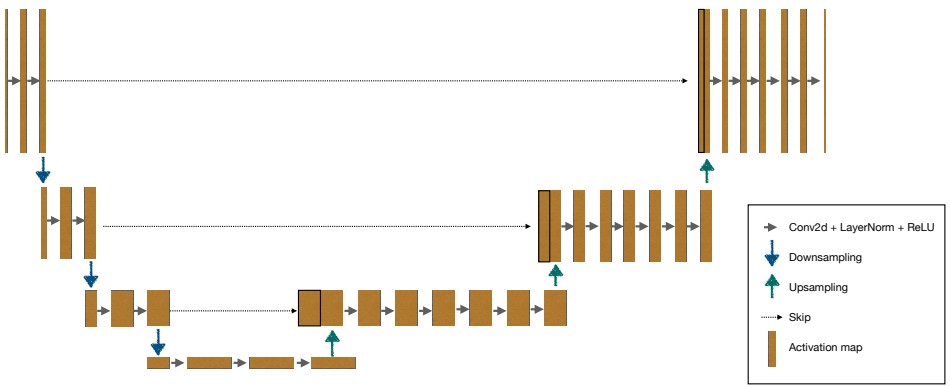

Figure 9: Fully convolutional UNet architecture used in our experiments.

## A.2  DATASETS

We show results from models trained on 4 public datasets:

- ImageNet64: down-sampled version of ImageNet data set (Deng et al., 2009) , $3 \times 64 \times 64$. The training set consists of $\sim 1.2m$ images and the validation set consists of $50k$ images, of objects, animals, scenes, etc. We did not use the class labels for training.

- LSUN-Bedroom dataset (Yu et al., 2015): down-sampled to $3 \times 80 \times 80$ images. We trained the model on a subset of images randomly selected. Training set size $\sim 300,000$ images.

- CelebA dataset (Liu et al., 2015): down-sampled to $3 \times 80 \times 80$ images. Training set size $\sim 200,000$ images.

- Texture dataset (collected by the authors, not published), cropped to $3 \times 80 \times 80$ images. Training set size $\sim 230,000$ images.

### A.3 TRAINING

The network was trained to minimize mean squared loss between output of the network and clean image, $\ell(\theta) = \mathbb{E}_{x,\sigma,z}\|x - \hat{x}(x_\sigma)\|^2$ , using Adam optimizer, with an initial learning rate of $0.001$ with a decay of a factor of 2 every 100 epochs. Total number of epochs was set to 1000. The size of each batch was 1024.

The standard deviation of noise was drawn randomly for each image from a $\frac{1}{\sqrt{\sigma}}$ distribution to emphasize the small noise levels, since we have observed that the it takes more epochs for the denoising MSE to plateau for small noise levels under uniform $\sigma$ distribution.

## B PROPERTIES OF REPRESENTATION

### B.1 REPRESENTATION SPARSITY

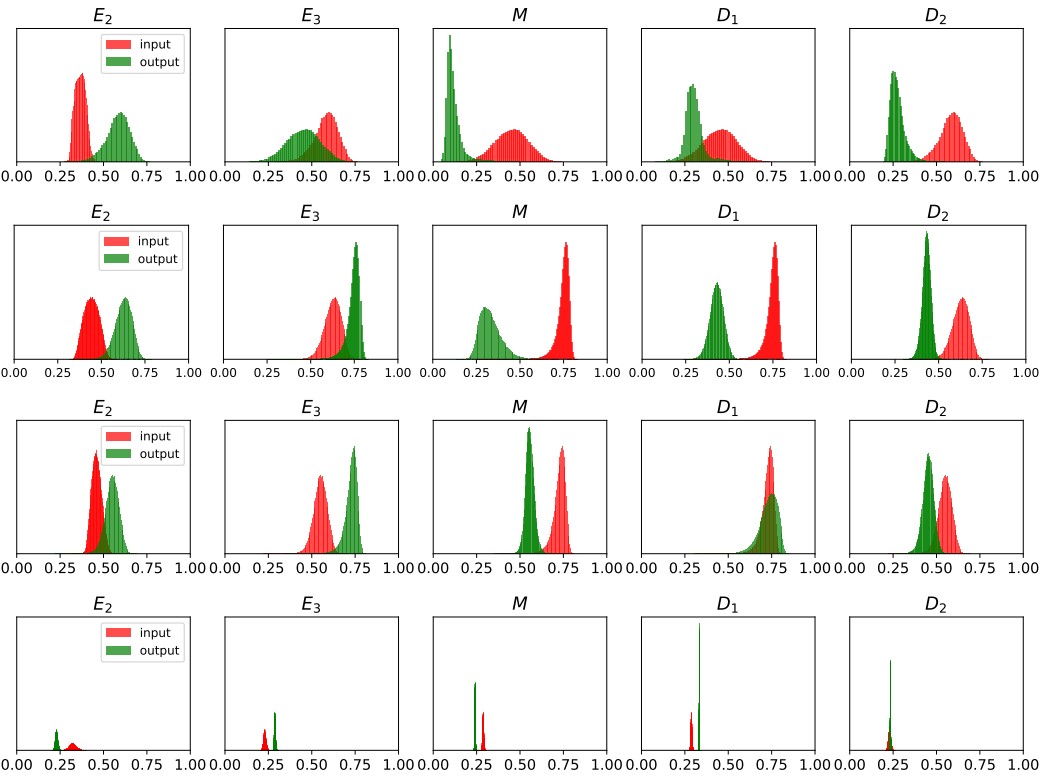

Figure 10: Change in sparsity for representations in models trained on **Texture**, **LSUN-Bedroom**, **CelebA** datasets, and an **untrained** randomly initialized model, from top to bottom row. For trained models, channel sparsity increases in the Middle block across dataset. See caption Figure 1 for detailed description. For an untrained model, tested on ImageNet dataset. The sparsity distributions are narrow and nearly identical across images. The input–output change in sparsity is minimal and directionally random.

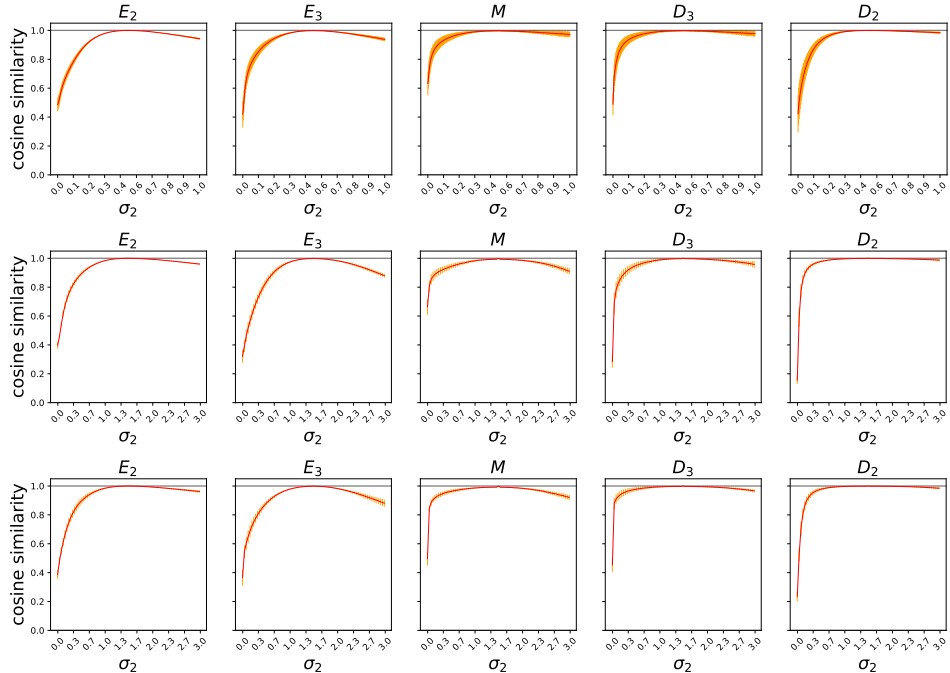

Figure 11: Stability of $\bar{a}$ across noise levels, for different network blocks trained on **Texture**, **LSUN-Bedroom**, and **CelebA** dataset. See caption of Figure 2 for detailed description.

## B.2 CHANNEL SELECTIVITY

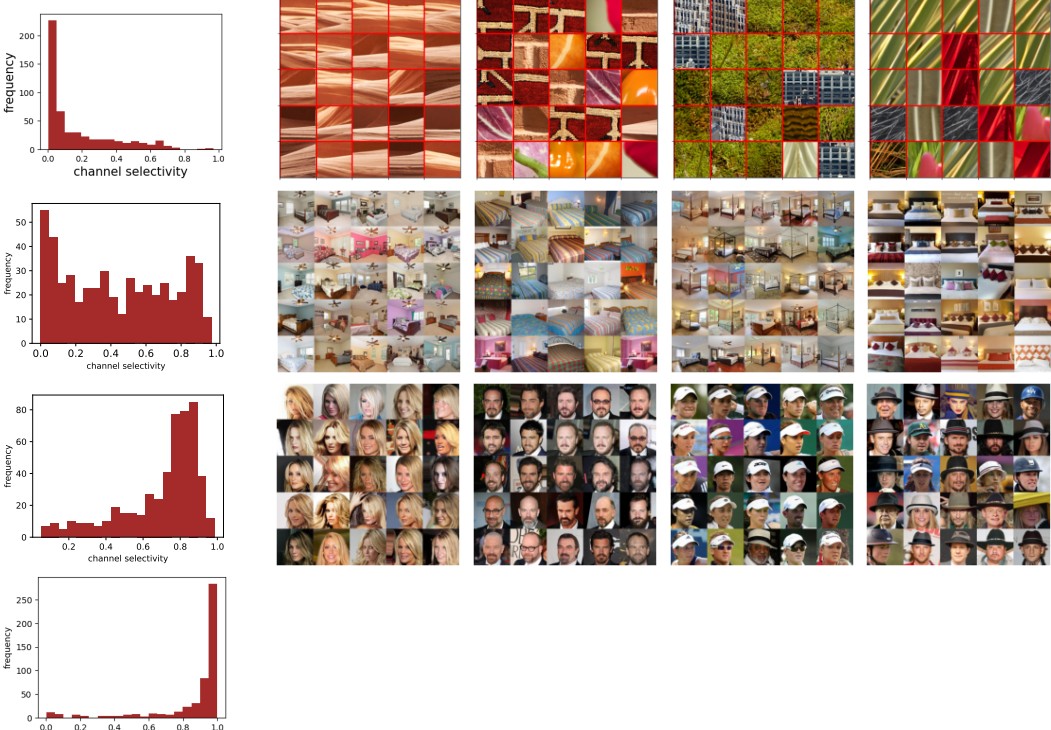

Figure 12: Channel selectivity for four models trained on Texture, LSUN-Bedroom, CelebA datasets, and an untrained model **Texture dataset:** Distribution of channel selectivity shows that the vast majority of channels have low PR, corresponding to channels that are highly specialized (and infrequently active), concentrated on the left. The panels show the set of images that maximally activate each of four specialized channels, revealing selectivity for different texture patterns. PR of channels from left to right for texture model: $0.013, 0.024, 0.027, 0.038$. **LSUN-Bedroom:** Channel selectivity is distributed more evenly with more common channels compared to the models trained on ImageNet and Texture datasets. This can be attributed to the presence of more common patterns and structures within the images of the dataset, as they all depict bedrooms. The PR of the four channels shown are $0.015, 0.053, 0.018, 0.36,$, from left to right. **CelebA:** Most channels in this model respond to the majority of images. This can be attributed to the fact that all images in the dataset share the global layout and coarser level structures since the faces are aligned and centered. The four channels shown here have PR of $0.29, 0.309, 0.36, 0.50$. **Untrained model**: Channels from an randomly initialized, untrained model show almost no selectivity to different images from the ImageNet dataset.

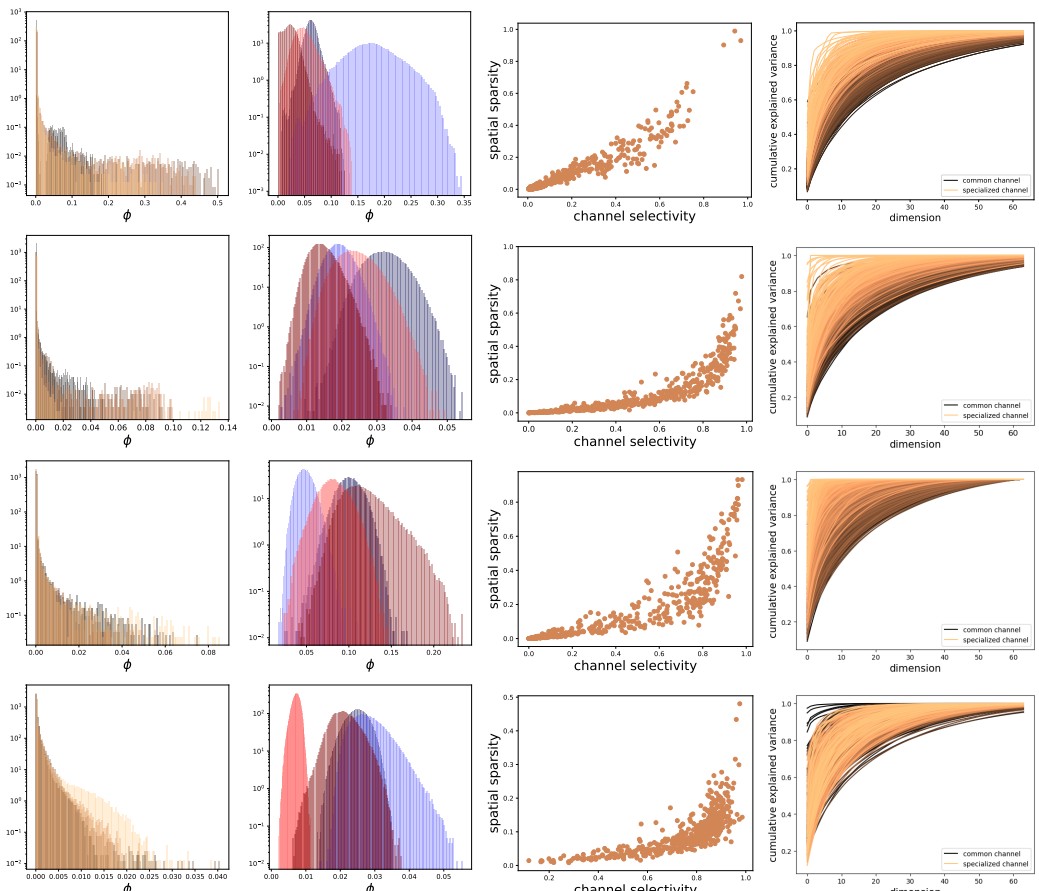

Figure 13: Channel selectivity predict signature statistical properties of the channels. From top to bottom row, models trained on **Texture dataset**, **LSUN-bedroom**, **ImageNet** and **CelebA**. From left to right: 1) marginal distribution of $\phi[i]$ in 4 specialized channels. 2) marginal distribution of $\phi[i]$ in 4 common channels. 3) channel selectivity is correlated with spatial sparsity within the channel. 4) The spatial variance of specialized channels is explained with fewer dimensions, measured by cumulative explained variance of PCA analysis

### B.3 SEMANTIC SIMILARITIES IN REPRESENTATION SPACE

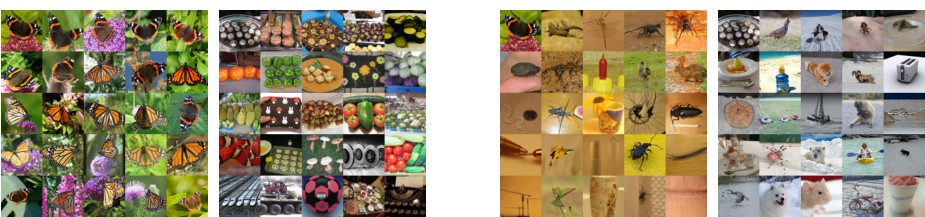

Figure 14: Nearby $\phi$'s correspond to images that are semantically similar. **Left:** Two sets of images, each showing a target image (upper left), and a set of images whose $\phi$'s are closest to that of the target image in terms of cosine similarity. For the butterfly image, proximity in latent space is aligned with the object identity. For the tray of food, proximity is aligned with images depicting collections of similar objects. **Middle:** Same as left, but showing images closest to the target in terms of cosine similarity in the pixel domain. **Right** Symmetrized KL divergence between the distributions of conditional samples, plotted against Euclidean distance in the representation, for randomly selected pairs of $\phi$ vectors.

**clustering.** We used the `python` implementation of K-Means clustering algorithm, from the `sklearn` package. Number of clusters, K, is set to 1000 following the number of human labeled classes in the dataset. Clustering at different noise levels leads to similar results, consistent with stability of $\phi$ over noise levels shown in Figure 2. This is not true when noise is very small, since $\phi$ collapses to zero at small noise levels. K-mean clustering algorithms are sensitive to initialization, so the assignments of images to clusters changes with initialization, but always leads to the same semantic grouping. Even initialization from the centroids of the pre-defined class labels results in similar clusters. This means that even when we give the algorithm a chance to cluster the images within the same class together, it pushes away from that and re-allocate the assignments such that the images are grouped together based on "the gist of the scene" as opposed to object identity.

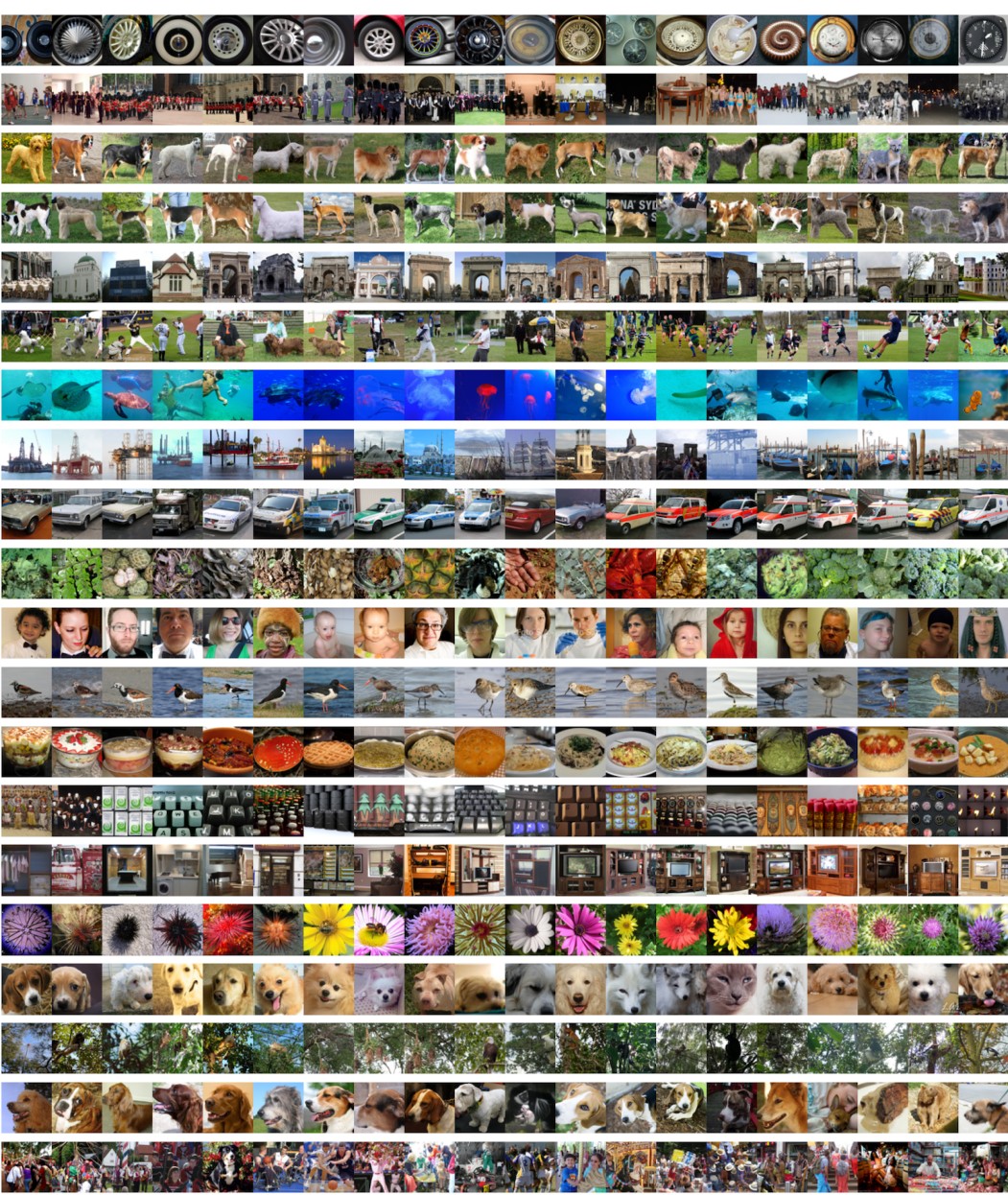

Figure 15: Continued from Figure 6. Random images from different clusters are shown in each row. Different cars with the same orientation are clustered together, but similar car brands with different orientations are assigned to different clusters. (see rows 2 and 3).

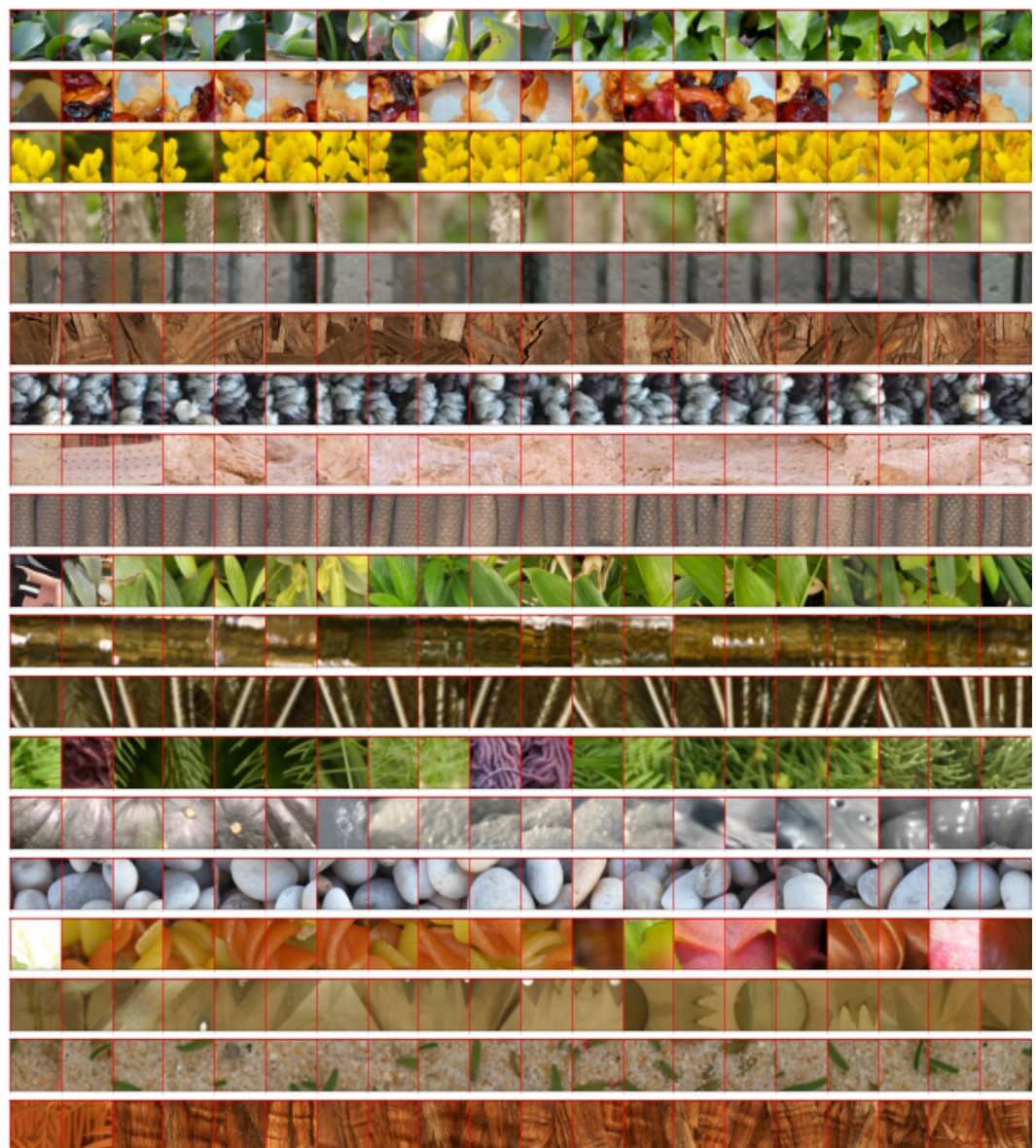

Figure 16: Random images from different clusters are shown in each row. Obtained from the model trained on **Texture dataset**.

## C  STOCHASTIC RECONSTRUCTION ALGORITHM

We build our sampling algorithm based on the algorithm in (Kadkhodaie and Simoncelli, 2021) which does not require the noise level and follows an adaptive step size schedule. Hence, this network does not take the noise variance as an input and is a blind denoiser. This setup has the advantage that simplifies the network, which is important for analysis of the internal layers of the model.

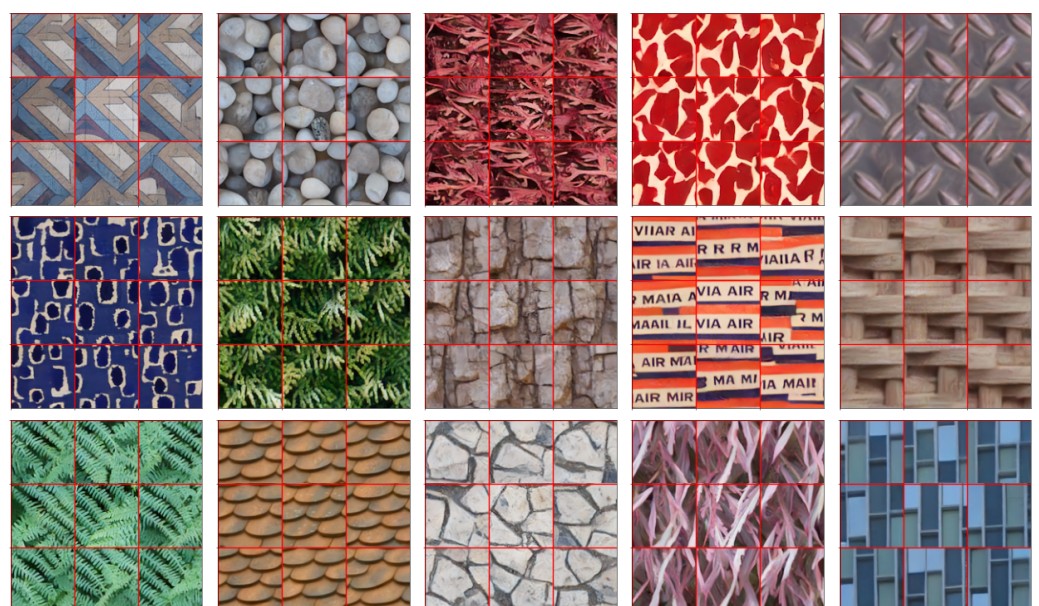

Figure 17: More samples from model trained on Texture dataset. See caption of Figure 7.

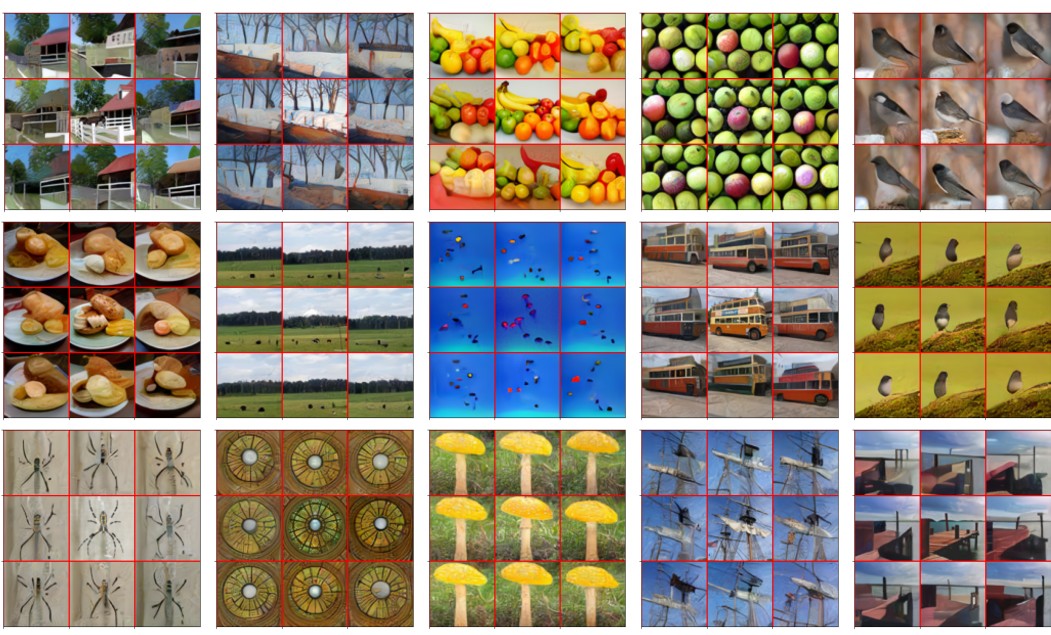

Figure 18: More samples from model trained on ImageNet64 dataset. The samples are visually similar to the target image at the center of each panel. Interestingly, the location of the large, long-ranging image patterns are persevered in the samples, while the fine structures and details are diverse in their location with respect to the target image. See caption of Figure 7.

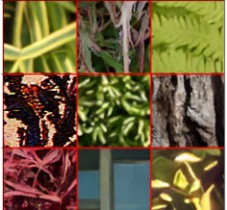 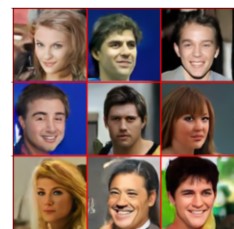 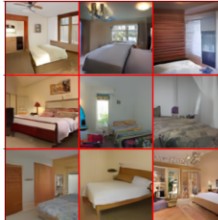 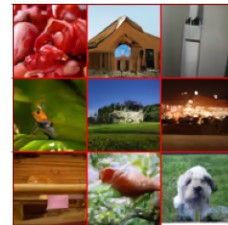

Figure 19: Samples generated unconditionally from models trained on Texture, CelebA, LSUN-Bedrooms, and ImageNet64 datasets respectively.

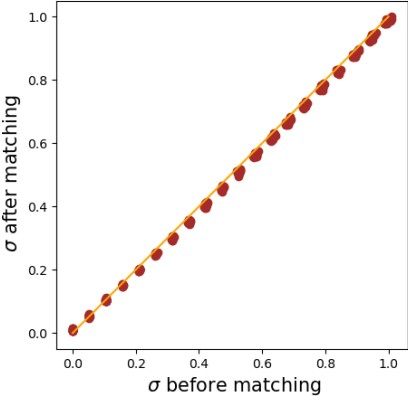

Figure 20: Matching does not change the noise level $\sigma$. This plots shows the standard deviation of noise on the image before and after the matching step, for a collection of random test images, at different input noise levels. $\sigma$ after matching is measured by the norm of score divided by the square root of ambient dimensionality, $\|s(x_\sigma)\|/\sqrt{n}$. This is an approximation of the strength of the remainder noise on the image by assuming a Gaussian prior.

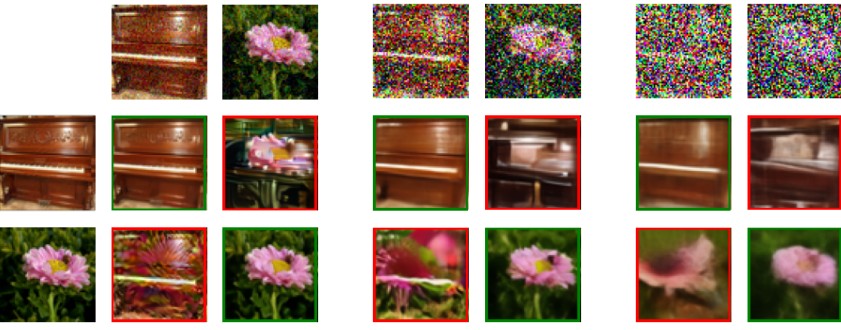

Figure 21: A pair of clean images, $x$ and $x'$, from ImageNet dataset is shown on the first column. The distance between the the two densities induced by the $\phi$'s obtained from these two images can be defined and computed using Equation (4). The first term in the integrand is the difference between two conditional scores: the score of $x_\sigma$ given $\phi$ and the score of $x_\sigma$ given $\phi'$. This difference is computed and integrated at all $\sigma$ levels (and symmetrized). For these two images, this difference is visualized in the green versus red boxes.

