# OpenReview forum: "Unconditional CNN denoisers contain sparse semantic representations of images"
_ICLR.cc/2026/Conference — Submitted to ICLR 2026_

### Official Review · Reviewer_GomQ · 2025-10-25

**Soundness:** 2
**Presentation:** 3
**Contribution:** 2
**Rating:** 2
**Confidence:** 3

**Summary:**

This paper investigates the representational properties of the channel-wise average vector extracted from the middle block of a denoising U-Net (or a score U-Net, following Tweedie’s formula). Extensive experiments demonstrate that this vector encodes perceptually meaningful image representations, which can be effectively leveraged for clustering. Finally, the Euclidean distances between these representations are exploited within a diffusion-based algorithm to generate variants of a conditioning image.

**Strengths:**

1. The intuition behind this paper draws inspiration from classical denoising techniques (such as DCT or wavelet denoising), where noisy signals are decomposed into concentrated (sparse) and distributed (dense) components.
2. The paper is well written and easy to follow.
3. Several representational properties of a denoising network were highlighted, including increase of sparsity through the channels, robustness of representation to noise, channel selectivity and clustering.

**Weaknesses:**

My overall impression is that, although the proposed algorithm is effective to generate variants of a conditionner image, the article may over-estimate the role played by the features of a denoising network. I recommend additional experiments to clarify if the effectiveness of the algorithm is dependent on it or if the "guidance step" could leverage a VGG-based perceptual loss instead (see questions below).

Here are some additional weaknesses:
1. The conclusions are drawn from only one type of denoising architecture, namely UNet (Ronneberger et al., 2015).
2. The use of the normalized participation ratio for measuring sparsity should be better justified.
3. The rational of Algorithm 1 should be better explained. For example, line 963, it is mentionned that the sampling algorithm is based on Kadkhodaie and Simoncelli, 2021 but without any explanations.
4. No code was provided.
5. DCT denoising reference [a]  is missing on line 146. t-SNE reference as well.
6. Line 245, PCA analysis is not provided.

[a] G. Yu and G. Sapiro. DCT image denoising: a simple and effective image denoising algorithm. Image Processing On Line, 2011.

**Questions:**

1. Can you draw the same conclusions (increase of sparsity through the channels, robustness of representation to noise, channel selectivity, clusters of images, etc) with an **untrained** UNet? What about other types of architecture such as DnCNN [b]? Or even networks trained for classification?

2. Is the paragraph on translation equivariance of UNets really useful? (Line 100 to 132). I do not understand what you are getting at. Would the analysis be that different with a true translation equivariant model (e.g., DnCNN [b] with circular padding). By the way, can you observe the same increase in sparsity throughout the 17 layers of DnCNN [b]?

3. What happens if only the "distance between representations" is minimized (for initialization we can consider a noisy version of the conditionner image)? (The algorithm would simply consist in repeating the "Guidance step" until convergence).

4. What happens if the "distance between representations" is replaced by a VGG-based perceptual loss [c] in your algorithm?

5. In Figure 3, what does it mean mathematically to "maximally activate" a channel? By the way, are the images in the panels you show in Figure 3 and 4 ranked by their degree of excitation, whatever that means?

6. Which loss was used exactly to train your networks? Is it the one from (1) which minimizes the MSE between the output of the network and the corresponding pure noise?

[b] K. Zhang et al. Beyond a gaussian denoiser: Residual learning of deep cnn for image denoising. IEEE TIP, 2017.

[c] J. Johnson et al. Perceptual losses for real-time style transfer and super-resolution. ECCV, 2016.

---

> ### Author Response · Authors · 2025-11-21
>
> Thank you for your comments.
>
> **About the proposed experiment:**
>
> Our reconstruction algorithm uses **only the encoder features of the denoising model itself** to guide sampling, so its behavior **necessarily** reflects **only** those features.. Replacing the guidance step with a VGG-based perceptual loss would rely on representations from an external network trained for classification, not on the denoiser’s representation that our work aims to interpret. Such an experiment would mix the influence of two unrelated feature spaces, making it difficult to attribute any effect to the denoiser’s features. In short, our setup is designed to isolate and analyze the representation learned by the denoising network; introducing an external perceptual loss would confound rather than clarify this analysis.
>
> **Other denoising architectures including DnCNN:**
>
> We chose to focus on **U-Net** for two main reasons.
> 1) **Relevance**: U-Net is by far the most widely used convolutional backbone in modern denoising and diffusion models. Understanding its internal mechanics is therefore of broad significance.
> 2) **Scientific scope**: Our goal is to study how and why this specific architecture works, not to perform an exhaustive comparison across models. Such comparative analyses would require extensive additional experiments and are beyond the scope of this paper.
>
> Regarding **other architectures** (e.g., DnCNN): denoising mappings are inherently **contractive**—they reduce noise amplitudes and thus increase sparsity in most directions of the feature space whenever the mean-squared error improves. This relationship between **sparsity and denoising** is a fundamental property of image restoration, not specific to a particular architecture. Therefore, we would expect a similar increase in sparsity somewhere in the DnCNN model, or any other effective denoising architecture.
>
> What differs across architectures is **where** in the network this contraction occurs, which requires detailed layer-wise analysis similar to the one we performed here. We agree that such a study would be valuable future work.
>
> Finally, while **DnCNN** was an influential early denoising architecture, it has been largely replaced in diffusion models due to its **limited receptive field (~40×40)** and lack of scalability to larger image resolutions. In contrast, U-Net scales efficiently and remains the architecture of choice in nearly all modern diffusion-based models.
>
> In summary, our analysis provides a **mechanistic understanding of the most widely used architecture in diffusion and denoising research**, and the mathematical principles we highlight apply broadly across architectures.
>
> **Participation ratio (PR):**
>
> The normalized participation ratio (PR) is a well-established measure of **effective dimensionality** and **soft sparsity** in representation learning, physics, and neuroscience (e.g. Gao et al., Neuron 2017). The squared ratio of $L_2$ and $L_1$ norms ranges from 1 (for a sparse vector with a single non-zero entry) to N (a dense vector filled with identical entries), providing a smooth, differentiable proxy for the number of active components in a vector—effectively **an $L_1$ relaxation of the  $L_0$ norm** (which counts the number of nonzero coefficients).
>
> We chose PR because it captures the distribution of activity energy across neurons rather than relying on an arbitrary threshold for zero activations, making it both robust and interpretable. The normalization ensures comparability across layers and scales.
>
> Given its extensive use as a measure of representational dimensionality and sparsity in the literature, we believe the choice is well justified.
>
> [Gao, Peiran, Eric Trautmann, Byron Yu, Gopal Santhanam, Stephen Ryu, Krishna Shenoy, and Surya Ganguli. "A theory of multineuronal dimensionality, dynamics and measurement." Neuron (2017)]
>
> **Choice of sampling algorithm:**
>
> We chose the sampling algorithm of Kadkhodaie & Simoncelli (2021) because it provides a simplified and analytically tractable setup, which is **crucial for studying the internal representations of the network**. This formulation reduces architectural complexity while preserving the essential dynamics of stochastic reconstruction, making it particularly suitable for the **interpretability** analysis that is the focus of our paper.
>
> There exist many variants of diffusion or sampling schedules, but our goal was not to benchmark them; rather, we required a clear and interpretable setting to analyze the model’s internal geometry.
>
> **Code:**
>
> We will release the full implementation, including training and analysis scripts, upon acceptance to ensure full reproducibility (stated in the original submission, in the reproducibility section). The code will also include instructions for reproducing all figures and experiments presented in the paper.

---

> ### Author Response · Authors · 2025-11-21
>
> ... continued:
>
> **Citation to t-SNE and DCT denoiser:**
>
> We added a citation  to the original t-SNE publication.
> The “classic denoiser” mentioned in the paper refers to the Fourier (Wiener) denoiser, which is a well-known baseline in the classic denoising literature.  The DCT denoiser is a variant, applied independently to local blocks of pixels (typically 8x8).
>
> **PCA result:**
>
> The PCA results (cumulative explained variance by components) are included in Figure 13. To make this clearer, we have updated the figure caption to explicitly mention “PCA.”
>
> **Q: Does sparsity appear in classification models?**
>
> While some forms of sparsity (e.g., feature selectivity, pruning-induced sparsity) can emerge in classification networks, the mechanism and interpretation are fundamentally different from those in score-based denoising models.
>
> In denoising networks, sparsity naturally increases because the learned mapping is contractive—it suppresses input noise and thus reduces the amplitude of feature activations along most directions of the signal space. This is a direct consequence of the denoising objective.
>
> In contrast, classification networks are trained to achieve discriminative separation between classes rather than to remove noise, so there is no inherent reason for a systematic increase in sparsity across layers. Any sparsity observed there would be driven by architectural or regularization choices (e.g., ReLU nonlinearity, dropout, L1 penalties) rather than by the training objective itself.
>
> Our study focuses on sparsity arising intrinsically from the score-based denoising loss, which has a clear mathematical interpretation as a contraction of the input distribution. Investigating analogous phenomena in classification models would address a different question and is beyond the scope of this work.
>
> **Q: Can you draw the same conclusions from an untrained model?**
>
> We thank the reviewer for this question, which led us to test it empirically.. We’ve now added quantitative results for an untrained U-Net in Appendix Figures 10 and 12. These results confirm that the untrained model behaves fundamentally differently from the trained denoiser.
>
> Specifically, in the untrained network the sparsity distributions are narrow and nearly identical across images, and the input–output change in sparsity is minimal and directionally random. Moreover, the activations show no image selectivity—the model responds similarly to all inputs.
>
> These observations demonstrate that the structured sparsity and selectivity patterns reported in the paper emerge as a consequence of training with the denoising loss, not from architectural bias or random initialization.
>
> **Q: why translation invariance and its breaking in UNet is relevant?**
>
> This paragraph explains why we have consistent localization of large scale features in synthesis. With a true translation-invariant model, location information cannot be obtained from the representation, and synthesis results should reflect this.
>
> **Q: Can you apply only the guidance step without the score step?**
>
> This procedure does not lead to denoising. As shown in Figures 8 and 20, the guidance step alone does not reduce the noise level—it only pushes samples within the set of high-probability images under $p_\sigma​$. When the feature distance $||\phi_{sample} - \phi_{target}||$ becomes zero, the process simply halts without approaching the manifold of clean images.
> The score-directed step is therefore essential: it explicitly moves samples toward regions of lower noise (the data manifold) by following the score (gradient of the log-density). Without it, the algorithm cannot achieve denoising or generate realistic reconstructions.
>
> **Q: What does maximally activating a channel mean?**
>
> For each channel i, we compute its activation value $\phi(x)[i]$ for all images x in the dataset and sort the images by this value. “Maximally activate” means selecting the images with the  highest activation in that channel—that is, the top-k inputs producing the largest $\phi(x)[i]$. The panels in Figures 3 and 4 therefore show the 25 images that elicit the strongest activations for each channel. These images are ranked.
>
> **Q: What is the objective?**
>
> We use the equivalent formulation that minimizes the mean-squared error (MSE) between the network output and the clean image rather than the pure noise. These two losses are mathematically equivalent (as expressed in eq 1). We chose this formulation because it makes the analysis and interpretation of feature activations more intuitive.

---

> > ### Comment · Reviewer_GomQ · 2025-11-26
> >
> > I thank the Authors for their response. However, in the absence of additional experiments with other networks, I remain convinced that the article may over-estimate the role played by the features of a denoising network. The rebuttal did not fully address my concerns, particularly regarding the purported universal increase of sparsity through the channels of the denoising network (contractive behavior), as well as the distinction from the features of a classification model for example. I therefore maintain my initial review. I keep my confidence score at 3.

---

### Official Review · Reviewer_G5Yd · 2025-10-30

**Soundness:** 1
**Presentation:** 2
**Contribution:** 1
**Rating:** 2
**Confidence:** 4

**Summary:**

This paper investigates the semantic latent space implicitly learned by denosing UNets trained for unconditional image generation using a Diffusion Model objective. First, the authors hypothesize that specific layers of UNet learn an explicit separation in feature space between the signal (image) and additive noise. The authors suggest a link between the sparsity in the input/output feature maps of each UNet block, quantified by the Participation Ratio (PR), and the denoising effect. Using this measure, the authors identify the feature map $\overline{a}\_4$, corresponding to the central block of Unet, as primarily responsible for denoising. Later $\phi(x\_\sigma)$, which appears to be defined as the mean of featuremap $\overline{a}\_4$ observed under different realizations of additive noise $\sigma$, is considered as a latent semantic representation of the input. In section 3, the authors propose a conditional generation strategy by alternating the DMP denoising step and a guidance step defined by minimizing $\phi(x\_\sigma)$ to the $\phi$ representation of a guidance image $x\_c$.

**Strengths:**

The work suggests an interesting perspective on the latent space structure of generative diffusion models. Analyzing feature-level implications of the denoising steps is a plausible direction to build a deeper understanding of the generative models low-level workings.

**Weaknesses:**

There are several clear limitations to this work.
First, the work is not properly framed: several prominent works have previously explored related concepts (Preechakul 2022, Kwon 2023, Wpstein 2023).

There are several scientific limitations to the proposed approach. Most of the contributions presented are neither linked to previous contributions nor supported by theoretical derivation nor sufficiently supported empirically by extensive experimental results.

This includes:

* The connection between representation sparsity and denoising. This connection is unclear and appears to be severely under-justified, and mainly accepted by assumption (ll 152).
* The hypothesis of the connection between sparsity reduction and noise suppression (l188-l197) is purely speculative.
* Similarly, the derivation of $\phi(x\_{\sigma})$ seems arbitrary and the notation is unclear. Is $\overline{a}\_4$ a function? Is $x$ the input of the UNet or of an intermediate layer? Is the noise realization added to the input of the denoiser (as in a typical diffusion model) or to the input of an intermediate layer?
* Conditional sampling: the concept of conditional sampling from Diffusion generative models trained unconditionally has been extensively investigated in previous works (Graikos et. al). Notably, such approaches suffer from off-manifold updates with respect to the Gaussian manifold of the latent distribution induced by the diffusion probabilistic objective.

Additional critical issues:
* The discussion of this work is based on image denoising concepts (sec 2.2, 2.3). This is not a problem in itself, but it creates confusion since it is not generally specified how these concepts relate to the generative diffusion framework. While it is true that the optimization goal of diffusion is denoising, the underlying probabilistic formulation cannot be ignored.

* Experimental analysis: Apart from a few qualitative observations, there is essentially no rigorous experimental analysis of the results obtained. This is particularly limiting given that the proposed work is mainly empirical. We strongly suggest that authors quantify their results (e.g., conditional generation) against benchmarks commonly used in similar works.

* Experimental setup: The description of the experimental setup used is extremely limited and unclear. A.3 (l652) does not make clear what the training objective is (diffusion, presumably?), and the constant references to a denoising model do not help to clarify matters.


References (non-exhaustive):\
Preechakul et. al, 2022, Diffusion Autoencoders: Toward a Meaningful and Decodable Representation\
Kwon et. al, 2023, DIFFUSION MODELS ALREADY HAVE A SEMANTIC LATENT SPACE\
Wpstein et. al, 2023, Diffusion Self-Guidance for Controllable Image Generation\
Graikos et. al, Conditional Generation from Unconditional Diffusion Models using Denoiser Representations\
He et. al, 2024, MANIFOLD PRESERVING GUIDED DIFFUSION

**Questions:**

* How was feature map sparsity, and consequently PR, chosen as a measure of the denoising effect? Are there any contributions in the field of deep learning to support this choice?

* How is the "mean of $\overline{a}$ across noise realizations" (l204) calculated in the definition of $\phi(x\_{\sigma})$?

* Were alternatives to $\phi(x\_{\sigma})$ evaluated in algorithm 2? (even just the raw features).

---

> ### Author Response · Authors · 2025-11-21
>
> **Framing of the paper:**
>
> All the cited works (Preechakul 2022, Kwon 2023, Epstein 2023) examine  **conditional** diffusion models, whereas our paper considers **unconditional** models. The difference is fundamental: conditional models are trained with external conditioning signals (e.g., class labels or text), while unconditional models must learn the data distribution solely from observed samples. This is akin to the difference between **unsupervised** and **supervised** representation learning. We have incorporated an additional paragraph (colored) in the introduction to emphasize this distinction.
>
> To our knowledge, only a handful of prior studies have analyzed **unconditional architectures**, and **these are all cited in the third paragraph of our paper**. Our contribution lies precisely in understanding the internal representations of this underexplored and foundational case.
>
> **Connection between sparsity and denoising:**
>
> We respectfully disagree. The relationship between **denoising and sparsity is a well-established result in signal processing literature**, not a speculative claim. Denoising mappings are inherently contractive: they suppress noise components , which corresponds to a reduction in the effective dimensionality or an increase in sparsity of the signal representation. This principle underlies classical frequency domain denoising (i.e., the Wiener filter), as well as more contemporary wavelet denoising formulations.
>
> **Notation:**
>
> We believe our notation follows **standard conventions in machine learning and representation learning**. The primary elements are::
>
> - $\phi$ is defined as the many-to-one mapping from the noisy input image $x_\sigma$ to the representation vector comprised of the spatial averages of activations of all channels of the last layer of the middle block of the U-Net.
> - $\bar{a}$ represents the spatially averaged activation (also shown in Figure 1) in either the input or output layer. The “bar” symbol is fairly standard for spatial average.
> - The noise variable $z$ is drawn from a standard normal distribution, and $\sigma$ denotes the noise standard deviation; noise is added to the input clean image $x$, consistent with standard denoising model formulations.
> - $\mathbb{E}$ denotes the expectation, estimated empirically by averaging across noise realizations.
>
>
> **Prior work on unconditional models (Graikos et al):**
>
> Graikos et al. use the term “representation” to refer to a different concept (not internal activations), therefore their paper addresses a **different concept**. They define **representation as the outputs of the generative process in pixel space** and examine  **representations along the sampling trajectory**. Their work is **not** about internal activations of the network, whereas our analysis examines **internal activations and feature representations within the denoising network itself**. Our use of the term representation refers specifically to these **learned internal features**, not to intermediate samples along the trajectory. So even though the title of the work looks similar to our work, it is in fact not related to analyzing the representation in its standard definition.
> We have added a footnote in the paper to emphasize this distinction and avoid potential confusion.
>
> **The use of the word and concept of denoising in diffusion modeling:**
>
> The connection between **denoising and diffusion** is at the heart of the diffusion modeling literature. Diffusion generative models rely on a pair of **noising–denoising operations**: the forward (diffusion) process progressively adds Gaussian noise, while the reverse (generative) process corresponds to **iterative partial denoising steps**.  A denoiser that minimizes the expected squared error computes the conditional mean of the posterior, which can be expressed in terms of the  score function $\nabla_x \log p_\sigma(x_\sigma)$.  Thus, the generative process is a form of “score ascent”.
> Since the focus of our work is not training or sampling but representation learning, we examine the properties of the denoising (score) network.  We don’t provide comprehensive background sections on the training or sampling, but reviews are now plentiful in the ML literature.
>
> **Benchmarking the stochastic reconstruction algorithm:**
>
> Our goal is **not to propose a new sampling method**, but to use the reconstruction procedure as a **tool** for analyzing the internal representations of diffusion models. For this reason, standard generative benchmarks are not directly applicable or informative for our purpose.
> To our knowledge, this is the  **first study that uses a model’s own internal activations to guide synthesis** and to probe its learned representation geometry. There are therefore no established quantitative benchmarks for this type of analysis.
> We emphasize that the qualitative results serve to illustrate and interpret representational behavior, not to claim sampling performance improvements.

---

> ### Author Response · Authors · 2025-11-21
>
> ... continued:
>
> **Objective:**
>
> The training objective is indeed the standard denoising loss used in diffusion models, which is provided in Equation (1). Specifically, we minimize the mean-squared error between the predicted and target clean images for additive Gaussian noise with variance sampled from a fixed schedule.We highlighted this in the training section of the appendix for emphasis.
>
> **Papers reviewer cited as prior work:**
>
> Again, there appears to be a **confusion about conditional and unconditional models (supervised vs unsupervised learning)**. The first three articles (Preechakul et. al, 2022; Kwon et. al, 2023; Epstein et. al, 2023) examine **conditional diffusion models**.  He et. al, 2024 uses an **external** off-the-shelf autoencoder (e.g., VQGAN) to steer a diffusion model. And Graikos et. al, defines “representation” as the intermediate sample in the sampling trajectory (one-shot denoising output applied to samples in pixel domain), which is quite different from what we study in our paper.  After extensive searching, we found only 5 papers that used representation of unconditional models.  All of these are cited and described in our original submission.
>
>
> **Is there literature on the relationship between denoising and sparsity?**
>
> Yes. There is **extensive literature** on this topic which goes back many decades. It is studied and used in denoising  methods based on wavelet thresholding, sparse coding, K-SVD. See second point under **Connection between sparsity and denoising**
>
> **How is the "mean of  across noise realizations" computed?**
>
> Expectation is approximated by averaging (across noise realizations).
>
> **Were alternatives to $\phi(x)$ evaluated in algorithm 2? (even just the raw features)**
>
> It is not clear to us what is being asked here. Can you elaborate on what you mean  by “raw features”? The definition and motivation for use of $\phi(x)$ comes from the observation that it is sparse: for the imageNet experiments, only about 20% of the channels are significantly activated.  This 20% carries semantic information about the image, as demonstrated throughout the paper.

---

> > ### Comment · Reviewer_G5Yd · 2025-11-26
> >
> > I thank the authors for their responses. In particular, I appreciate the effort made to clarify the positioning of the proposed work in a purely unconditional context, which clearly distinguishes it from some of the references suggested.
> >
> > Unfortunately, some fundamental aspects remain unclear.
> >
> > **Relationship between denoising and sparsity.**
> >
> > I agree with the authors that this is a well-known result in the field of signal processing. The fact that the same principle applies to deep denoisers, and in particular to denoisers trained with a DPM objective, is still a hypothesis. Other reviewers also raised a related concern about this property emerging as an architectural bias rather than from the denoising objective.
> >
> >
> > **Alternatives to $\phi(x)$.**
> >
> > What I meant by "raw features" is that a baseline can be established by just leveraging a feature representation of the conditioner image (i.,e $\overline{a}_4(x_t^c)$) in the minimization problem in Algorithm 2. Is the expectation across noise realizations especially beneficial?

---

### Official Review · Reviewer_a9ev · 2025-10-30

**Soundness:** 4
**Presentation:** 4
**Contribution:** 3
**Rating:** 4
**Confidence:** 5

**Summary:**

This paper studies internal representations in diffusion models by examining spatially averaged activations in the UNet denoiser. The authors argue that these activations form a sparse latent code that captures high-level structural information, and they demonstrate that manipulating this code enables controllable sampling. The empirical exploration is interesting and the presentation is clear, with visualizations that support the central narrative.
I find the direction compelling. Understanding what diffusion models encode internally is an important challenge and this paper takes a meaningful step toward that goal.

**Strengths:**

* Innovative perspective on internal representations in diffusion denoisers.
* Well structured and clearly written.
* Intuitive visualizations that help explain the emergence of latent structure.
* Creative approach to conditioning sampling through internal activations.

**Weaknesses:**

I enjoyed the paper and I am genuinely interested in the mechanism behind the observations. I want to raise an alternative interpretation that might be worth discussing, though I am not fully confident in it and would appreciate the authors' take.
My current intuition is that some of the observed structure could also arise from well-known properties of convolutional UNet architectures rather than being specific to diffusion learning. In particular:
* UNets naturally encourage preservation of coarse structural information (“scene gist”) due to local receptive fields and hierarchical feature aggregation (early convolutional layers tend to capture low-level edges and textures, while deeper layers integrate spatial context, producing global shape descriptors) and skip connections (the encoder-decoder structure with long skip connections allows high-resolution spatial structure to flow directly to the decoder, which biases the network toward retaining global geometric configuration even under noise). This composition is known to support coarse scene layout understanding in CNNs even without explicitly modeling semantic content. As a result, the emergence of “gist-like” representations, large-scale silhouettes, dominant contours, and coarse texture organization, might be a natural consequence of convolutional structure rather than specifically of the diffusion learning objective.

* Deeper layers in UNets typically have more channels. Averaging many activation channels can reduce variance roughly as $\frac{\sigma^2}{n}$. The observation that the deeper block has lower average activation magnitude might therefore reflect channel scaling in the architecture rather than semantic sparsity created by diffusion training.

Again, my intention is not to undermine the paper. I am simply curious whether the authors believe these architectural factors alone could explain the results, or whether diffusion training plays a more essential role. A comparison against a supervised denoiser or a randomly initialized UNet could clarify this point.

**Questions:**

Already addressed in the "Weakness" section.

---

> ### Author Response · Authors · 2025-11-21
>
> We thank the reviewer for this insightful comment.
>
> **Q: Could the emergent properties be due to the architecture not the loss?**
>
> The U-Net architecture indeed **provides the expressivity** to preserve image structure and learn hierarchical representations—it defines a function class that can capture such relationships. However, realizing these structures in practice requires **learning through optimization of the denoising loss**.
> To verify that these representational patterns arise purely from architectural bias, we have now added quantitative results for an **untrained U-Net** in Appendix Figures 10 and 12. The untrained model behaves fundamentally differently:
> - The **sparsity distributions** are narrow and nearly identical across images.
> - The **input–output change in sparsity** is minimal and directionally random.
> - Most importantly, the **activations show no image selectivity**, responding similarly to all inputs.
>
>
> These observations demonstrate that the structured sparsity and selectivity reported in the paper **emerge as a consequence of training with the denoising objective**, not from architectural bias or random initialization.
>
>
> **Q: Could the emergent properties be due to merely having more channels in the deeper layers?**
>
> Our analysis does **not rely on activation magnitude** but on a **normalized sparsity measure (the participation ratio)** that is invariant to rescaling or change in dimensionality of the phi vectors. If the lower average activation magnitude were merely due to having more channels, the variance would distribute more uniformly across channels, resulting in **smaller but uniformly spread activations**—which would yield a high participation ratio. In contrast, what we observe is that participation ratios decrease with network depth, indicating that activity **concentrates in a smaller subset of channels**.

---

> > ### Comment · Reviewer_a9ev · 2025-11-26
> > **Thank you for clarifying with experiments**
> >
> > The new experiment on untrained architecture partially answer my raised questions. However, it still could be that the property you observe is just a consequence of the UNet architecture, but it must be trained (with any loss, not necessarily the denoising objective).
> > In particular, I believe that if you repeat the same experiments with any pre-trained UNet from literature trained on the task of image reconstruction, you will observe similar behavior.
> >
> > I am satisfied with the reply on my question about the number of channels.
> > I will raise my rating to 6.

---

### Official Review · Reviewer_xFVX · 2025-11-03

**Soundness:** 3
**Presentation:** 3
**Contribution:** 3
**Rating:** 6
**Confidence:** 3

**Summary:**

- The main finding of this paper is that unconditional denoiser models (U-Net type), when trained on image datasets, exhibit at the top middle block a certain level of semantic understanding such as structure, pose, shape, texture, and color, although their human-level semantic class understanding remains limited.
- The paper motivates this finding from the perspective of a denoiser that learns to separate signal from noise (analogous to a Wiener process). This leads to their proposed sparsity measure, which can be used to characterize each learned feature at the population level.
    - The participation ratio (PR) is introduced to measure how "sparse" is a spatially averaged feature of the U-Net.
        - High participation => dense => the latent lives in a high dimensional space.
        - Low participation => sparse => the latent lives in a low dimensional space.
        - Naturally, noise is high dimensional and image manifold / semantic information is low dimensional.
    - The paper shows that the U-Net encoder layers might be recognizing the noise as measured by the increased PR from their output compared to input. However, almost always at the middle block, the PR is decreased from their output compared to the input suggesting that the layer starts subtracting noise from the input resulting in signals. This seems to be the case for the subsequent decoder layers as well.
        - This finding motivates looking at the signals precisely at the top middle layer.
    - This PR (at the top middle layer) when aggregated over the whole dataset can be used to understand the features live in the space by looking when / how often / and where they have high participations. This is the main mechanism that the paper uses to discover / analyze semantic features in the denoiser models.
- The paper provides substantial qualitative evidence based on these measures, which I find convincing and that successfully changed my perception of the quality of denoiser model features.

**Strengths:**

*Note: I am reviewing this paper as a non-veteran reader of model representation studies. I may find some findings more interesting than they should be due to limited exposure.*

- While the model’s semantic class understanding is still limited—and not entirely unexpected as partially shown in DIFT (Tang 2023) and Mukhopadhyay (2023)—the paper nonetheless provides a convincing qualitative story and evidence.
- The overall qualitative evidence and analysis genuinely changed my perspective on the closest relative, diffusion models, representations.
    - I used to believe that diffusion models, being trained on pixel losses, must remain largely grounded in pixel distances despite their hierarchical representations. The degree of semanticity expressed—especially in Figure 4 (people) and Figure 15 (people and harbor)—exceeded my expectations. I am happy to say that I learned something new from this paper.
- I find the PR metric (which I believe is motivated by the Wiener process) simple yet insightful, and it is used very effectively in this paper. The subsequent φ values are also ingenious as a means of identifying feature subspaces. I find both novel.
- I find Figure 13 particularly insightful. It supports the claim (Line 243–245): “The marginal distribution of φ values in specialized channels is heavy-tailed, but common channels are closer to Gaussian.”
    - This finding is interesting for two reasons:
        - The confirmation that specialized channels follow a heavy-tailed distribution, while not surprising, is still valuable to know.
        - More intriguingly, the observation that common channels have Gaussian-shaped distributions raises deeper questions.
            - Beyond representation understanding, what does this imply about the dataset and learning process itself?
            - I cannot quite wrap my head around the Gaussian-ness, but if Gaussian distributions are narrow, does this suggest that features for common concepts can be captured with far fewer data points—perhaps following a faster scaling law than power-law regimes? Then, the remaining learning effort would focus on fitting the long tail of specialized concepts (which gives rise to the overall power-law scaling behavior).
- I like the idea of self-representation-guidance for visualization, which could be a useful tool for interpreting model representations (though its use in this paper feels somewhat limited; see Weaknesses).
- The paper is generous with clear visualizations and qualitative results throughout, including the appendix.

**Weaknesses:**

- I feel that the term *“semantic”* in the title might be a bit overclaimed, as the representations seem to prioritize shape, pose, and texture over human-defined semantics. The t-SNE plot (Figure 6, right) for human-labeled classes shows no clear separation of clusters, suggesting that the learned representation is still not strongly “semantic.”
    - To be clear, I understand that the model is unsupervised and should not be expected to produce human-level semantic groupings. My concern is purely about the choice of wording.
    - One question here: it is unclear how the t-SNE for human-labeled object classes was obtained. Why does this space look so different from the t-SNE plot on the left? I expected it to be the same, just colored differently by human labels.
    - Also, it's not clear how many clusters for K-means was chosen in the paper?
- Prior works on diffusion representations—such as DIFT (Tang 2023) and Mukhopadhyay (2023)—provided some quantitative analyses, which I find crucial. Assessing how well a model’s representation captures semantics is difficult to do via visualization alone. For example, in Figure 15, most rows could be explained by color and texture, which are not particularly “high level.” Including a quantitative task (that could be compared across models) would provide a clearer sense of the representation’s abstraction level.
- I find Figure 7 less insightful because most of the eight images are near duplicates of the conditional image. A more informative analysis would control for specific aspects or features—for instance, fixing a concept believed to represent a “wheel” and testing whether the generated images consistently depict wheels across scenes or poses.
- I don’t find the argument in Figure 1—that “the middle and decoder blocks exhibit increases in sparsity (i.e., reduction in PR)”—to be strongly supported, as Figure 10 tells a more ambiguous story (especially since the texture and CelebA datasets don’t seem to agree).
- Certain plots and analyses are hard to follow, such as Figure 6 (top-left ratio plot) and Figure 8. In general, I find the paper hard to follow at times. Maybe, due to my limited experience with this type of paper. The authors might help the readers of my type by making sure that the motivation / goal / take away message of each section is clear.
- While Section 3 is mostly easy to follow, I don’t find the finding in Lines 417–419 (“We show that Euclidean distances between φ’s are correlated with distances between the conditional densities they induce”) particularly valuable to the paper’s narrative. Even if true, it’s unclear what insight it offers—does it justify Algorithm 1’s correctness, or provide interpretability?

**Questions:**

While the finding here is not totally unheard of, I'm leaning toward a weak accept of this paper as it provides finding that confirm and in some cases (for me) exceed the expectations.

However, I still do have some skepticisms and questions::
- How well other "more primitive features" such as RGBs of 32 x 32 image, PCAs perform on the analyses in the paper (including K-Means in particular).
    - How for the other types of representations: MAE, DINO.
    - This will help the reader understand better how much better are the denoiser features.
- Since the denoisers are very similar to diffusion models which have t conditioned, it's natural to ask how do the features compare? Are they as robust (to the noise level) as the denoisers (I expect less robust to noise levels from the diffusion models).

In addition to that, I think the paper will have more values with:
- More quantitative semantic measures of the representations and comparisons against other models' representations.
- Realizing the promise of self-guidance generation to visualize what the model understands about a "concept" (not a whole picture).
- More discussion on (which may increase the reach and impact of this paper): What does it mean for common channels to have a Gaussian-shaped distribution?
    - The authors may or may not go into: Beyond representation understanding, what does this imply for the dataset and learning process? If Gaussian distributions are narrow, does this mean that we can capture common features with fewer data points and faster scaling laws than power laws?

---

> ### Author Response · Authors · 2025-11-21
>
> Thanks for your thorough review and thoughtful comments and suggestions, which helped us to improve the paper. The paper is also updated with **colored text** so the changes are easy to track.
>
> **The use of the term “Semantic” :**
>
> We used the term “semantic” to describe the grouping of images according to model subspaces (i.e., Figures 14, and 6/15/16). We agree with you that in the more specific sense (i.e. capturing object identity) the word “semantic” does not apply here. In fact, as you mentioned, we too don’t expect the model without supervision to capture object identity.  But we are quite surprised that the model can go as  far as it does, creating high-level concepts in the latent space, from the very low-level task of denoising, with no assistance from labels, augmentations, or auxiliary conditioning networks. Hence, the use of the term semantic here points to a broader sense as in the high-level abstraction as opposed to the low-level perceptual or pixel domain features.
>
> **t-SNE plot clarification (fig 6):**
>
> Thank you  for pointing this out. We modified the plot (see the updated paper) to address the question. The previous plot was showing 10 randomly selected clusters and 10 randomly selected classes, hence the two t-SNE plots were depicting different subsets of images. We have now **updated the figure and caption**: we show the same images in two plots, color-coded by cluster (left) and class (right).
> For K-means, the number of clusters, K, is set to **1000**, following the number of labels. We updated the text to clarify this.
>
> **D-prime for measuring distribution separation (fig 6):**
>
> We updated the caption and text  to clarify the upper left plot, which **quantifies the separation between image clusters**. Each point corresponds to a pair of clusters.  The y-axis shows the distance between their centroids. The x-axis shows the average standard deviation of the clusters, measured along the line that connect the two centroids.  The ratio of the two  coordinates is  “d-prime”, a standard measure of separability  in Signal Detection theory (larger = more separable). The left and right plots show clusters are well-separated, while object classes are more overlapping.
>
> **Alternative algorithms to steer towards “concepts”:**
>
> Thank you for this thoughtful suggestion. First, we note that **Figure 7** shows that the representation $\phi$ captures a large portion of the image structure. The high quality and low diversity among reconstructions supports this, demonstrating that the 512-dimensional $\phi$ vector encodes substantial image-specific information.
> We agree that a more controlled manipulation of individual features would yield valuable additional insights. In our preliminary experiments, we found that adjusting the amplitude of specific channels (for example, one associated with flower petals) led to systematic changes in the synthesized images, such as producing flowers with a greater number of petals. We plan to extend this line of analysis in future work to more systematically probe the **consistency, interpretability, and controllability of concept-level representations**.
>
>
> **What does the sparsity and selectivity indicate about other dataset?**
>
> (This point is also related to a question asked under the Strengths).
> Change in sparsity and selectivity depends on the underlying distribution, so they can be used to reveal statistical and geometric differences in the underlying probability density of the dataset. For example, CelebA dataset, which consists of aligned and centered faces, results in the least sparse and least selective representation. This makes sense because at the high level most images contain very similar large features, resulting in less selectivity and less sparsity. It means the union of subspaces in fact contains only a few subspaces, and they are not so low-dimensional. On the other hand, the images in the texture dataset have almost nothing in common in their high level large scale features, resulting in high levels of selectivity and sparsity. Most  channels in the former case have Gaussian marginals for their spatial averages, while in the latter case, most channels have highly sparse marginals.
>
>
> **What does the sparsity and selectivity indicate about learning ?**
>
> This is a very interesting question and your proposed intuition is consistent with the literature of learning: Low frequencies are learned first (since they have higher magnitude so they drive the loss) and higher frequencies are learned later in the training. Our frequency analysis shows that common channels generally capture lower frequencies while specialized channels capture higher frequencies. This implies the ordering you proposed in learning.
> In this work, we focused on the learned representation as opposed to the learning process, but following the evolution of selectivity and marginals in the channels could be a fruitful direction in future studies.

---

> ### Author Response · Authors · 2025-11-21
>
> ... continued:
>
> **The point of correlating distance metrics (section 3):**
>
> The purpose of this section is to make precise the notation of the similarity introduced as distance in $\phi$ space. One way of making it more precise is to show it is tightly connected with an established distance metric. The distance metric we chose is a symmetrized KL divergence that measures distance between densities induced by (conditioned on) $\phi$. This doesn’t quantify the level of abstraction of $\phi$ (we need human experiments for that), but it shows that $\phi$ is a proper embedding of the conditional density space.

---

> ### Author Response · Authors · 2025-11-21
>
> ... continued:
>
> **Robustness to noise in models with t-conditioning**
>
> It is possible  that the representation becomes more sensitive to noise level with explicit t-conditioning. However, we tend to believe that would be unlikely. Previous work has shown that explicit t-conditioning is not necessary to achieve good performance [Kadkhodaie & Simoncelli 2020 , Sun 2025]. At the same time removing t-conditioning makes the architecture simpler and more amenable to analysis. The fact that t-conditioning does not influence performance, implies that it doesn’t significantly change robustness of the representation to the noise level.
>
> Sun, Qiao, et al. "Is Noise Conditioning Necessary for Denoising Generative Models?." arXiv preprint arXiv:2502.13129 (2025).
>
>
> **Quantitative measurement of level of abstraction in the latent space:**
>
> We agree that developing a quantitative measure of the level of abstraction is an essential step. However, we view this as a future direction, primarily because a proper characterization of the semantic content of this representational space will likely require **human perceptual experiments**, similar to those used to study the “gist of a scene” (e.g., Oliva & Torralba, 2001).
> The main tool in the literature for measuring abstraction at the moment is linear classification of the representation. However, we already know from Fig 6 the clusters only **partially** overlap with human-labeled categories. This raises the question: what exactly do these clusters capture? It seems that they represent an intermediate form of information, lying **between object identity and the spatial layout of the scene**. Elaborating this is important, but would involve substantial additional work and is best suited for a separate paper.
>
> 1) **Classification :**
> Mukhopadhyay (2023) and the few other papers we found on the representation in unconditional diffusion models use **classification** as a downstream task to measure the abstraction or usefulness of the representation. That provides **a proof of concept** that there indeed **exists** a semantic representation in the model. **Importantly, their results show that a linear classifier head accuracy is roughly 20% below their best non-linear classifier head. This indicates that the natural grouping that arises from training is not fully aligned with class-labels.**
> Building on these results, we set out a different goal: to **understand and interpret** this representation and its properties (e.g. sparsity and selectivity).
> Internally, we trained **a linear head classifier** on our representation $\phi$, which resulted in **~40% accuracy** on ImageNet. This is far above linear classification in pixel domain **($~.01%$ accuracy)**, but far below the state of the art. This result and also visual inspection of the images within each cluster shows that low-level features like  color and texture cannot explain the similarity in representation space. Mukhopadhyay (2023) and the other papers cited in our submission, report higher classification accuracy, mainly because their goal is to **optimize** for classification result: they all use intensive **hyper-parameter grid search** for 1) optimal noise level, 2) optimal block 3) optimal average pooling size for classification, **tuned for different dataset**. The closest setting we found in Mukhopadhyay (2023) to ours was time (related to noise level)=90, block number=19 and pooling filter=8x8 with an accuracy of **55%**. The gap between our classification and theirs can be explained by the gap in architecture size (theirs ~500m parameters with 1024 channels in middle block vs. ours ~13m parameters with 512 channels in middle block), and the lack of hyper-parameter refinement in ours.
> In short, these prior papers show the **existence**  of a semantic representation. Our goal in this paper is to build on these results, not by improving accuracy,  but by **explaining them from a mechanistic interpretability perspective**. We updated the intro to make this point clearer. The novelty of our work is to make the representation explicit and reveal its geometric and statistical properties.
>
> 2) **Feature correspondence (PCK):**
> Thank you for pointing us to (Tang 2023) paper (DIFT) which we had missed. This work uses PCK, which is a an effective method of quantifying abstraction of representations. Here, representation is defined as the **entire feature map in all channels** in a layer (much higher dimensional than the input image), which is necessary for conducting PCK. Our representation (phi) has only 512 dimensions - far smaller  than the input image because of the **spatial averaging (global pooling).**   But that means it does not have access to the spatial feature maps. As a result, PCK cannot be applied to our representation.

---

### Author Response · Authors · 2025-11-21
**Global Comment**

**The primary goal of our paper is geometric and mechanistic interpretability:**

- Our purpose is to reveal, understand and characterize the representation that arises solely from a score estimation (or equivalently, denoising) loss, without use of  labelled data, augmentation, regularization, or auxiliary conditioning networks.
- **The transformation of a union of manifolds to a union of subspaces** has been a gold standard in representation learning (e.g. Buchanan 2025, DiCarlo 2007). We are showing **for the first time** that an unconditional diffusion model can achieve this, providing a novel form of geometric and mechanistic interpretability.
- The representation reveals a level of semantic abstraction that is surprising, given that the model is fully unconditional. As far as we are aware, the closest results in the literature that express this type of semantic grouping are those describing the “gist of a scene” (Oliva, 2005).
- We do *not* suggest that  this representation achieves competitive performance on classification or other downstream tasks. On the contrary, we show that class labels do **not** arise from denoising (Figure 6).
- While our paper does not fit neatly into either of the two dominant categories of validation (performance benchmarks vs. rigorous theoretical proofs) it nevertheless contributes to the scientific understanding of deep networks. We believe that this, although harder to evaluate, is important to the long-term goal of building a theory of deep learning.

**Distinction between unconditional and conditional models:**
- The vast majority of prior work on representation in diffusion models has used **conditionally** trained models. The top-down information flow arising from the conditioning information fundamentally changes the nature of representation. Here, we study unconditionally trained models, which do not have access to any explicit information about classes (through labels, augmentation, or text-based conditioning). **The distinction between unconditional and conditional models is fundamental**, analogous to the distinction between unsupervised and supervised/self supervised representation learning. To emphasize this, we used the term "Unconditional" in the title, abstract and throughout the text. . It is way more surprising to observe the emergence of semantic representation without semantic information included in the training.

Buchanan, Sam and Pai, Druv and Wang, Peng and Ma, Yi “Learning Deep Representations of Data Distributions”, (2025)

DiCarlo, James J., and David D. Cox. "Untangling invariant object recognition." Trends in cognitive sciences 11.8 (2007): 333-341

Oliva, Aude. “Gist of the scene”, Chapter 41 of Neurobiology of Attention, (2005): 251-256.

---

### Meta-Review · Area_Chair_iD6W · 2026-01-08

**Summary:**

The authors investigate the internal representations learned by unconditional diffusion-model. The main claim is that these models learn meaningful semantic structure, which arises from a denoising objective. A contribution is the Participation Ratio (PR) as a way to measure sparsity. Experiments are conducted on a UNet.

The initial recommendations for this manuscript were 1 marginally above acceptance threshold  (xFVX), 1 marginally below the acceptance threshold (a9ev), and 2 reject (G5Yd, GomQ). The reviewers appreciate the intuition of denoising from which the authors drew inspiration. However, there were a number of concerns that were raised by the reviewers including:

(1) overclaim in use of the term "semantic". The t-SNE plot (Figure 6, right) for human-labeled classes shows no clear separation of clusters (xFVX); (2) lack of quantitative assessment of grouping (xFVX);
(3) Fig. 7 being self-fulfillling as it near duplicates of the conditional image are used. Need analysis controlling for specific aspects or features (xFVX); (4) Fig. 1 claim of "the middle and decoder blocks exhibit increases in sparsity" is not well supported (xFVX); (5) unclear presentation, e.g., certain plots and analyses, are hard to follow (xFVX); (6) unclear what insight is offered by showing Euclidean distances correlating to distance between densities (xFVX); (7) comparison with features from other models (MAE, DINO), particularly t-conditioned denoisers, and sensitivity to noise (xFVX); (8) possible explanation of the "scene gist" due to architectural factors rather than denoising objective. (a9ev, GomQ); (9) the work is not properly framed (G5Yd); (10) unclear connection between representation sparsity and denoising, claim appears to be severely under-justified, speculative, and mainly accepted by assumption (G5Yd); (11) unclear derivation and notation (G5Yd); (12) connection to probabilistic formulation of diffusion is ignored (G5Yd); (13) unclear experimental set up and lack of rigorous experimental analysis, work is mainly empirical (G5Yd, GomQ); (14) conclusions are only drawn from one type of denoising architecture, specifically UNet (GomQ); (15) justification for normalized participation ratio as a measure of sparsity (GomQ); (16) lack of rational for Algo. 1  (GomQ).

While the authors was able to address some of the points raised by reviewers. There were several critical concern ins substantiating the claims made in the paper, such as the conclusions drawn being relevant broadly across architectures despite having only examining the UNet, the missing analyses and experiments, and unclear presentation. The AC recommends the authors to consider the outstanding points and incorporate the feedback and materials presented in the rebuttal, which would improve the next revision of the manuscript.

**Reviewer Concerns:**

The authors posted a rebuttal to address the following points:

(1) overclaim in use of the term "semantic". The t-SNE plot (Figure 6, right) for human-labeled classes shows no clear separation of clusters (xFVX):

The authors noted that the term "semantic" is used to describe the grouping of images according to model subspaces. The AC agrees with the reviewer that the use of semantic, especially for audiences of the venue, is misleading and advises the authors to choose a more fitting term. The authors corrected Figure 6. This point is partially addressed.

(2) lack of quantitative assessment of grouping (xFVX):

The authors added a plot to quantify separation between image clusters. This point is point addressed.

(3) Fig. 7 being self-fulfillling as it near duplicates of the conditional image are used. Need analysis controlling for specific aspects or features (xFVX):

While the authors mentioned that they had preliminary experiments that involve adjusting specific channels to accentuated certain features. However, they did not provide such experiments or the requested experiment from the reviewer in the rebuttal. This point was not addressed.

(4) Fig. 1 claim of "the middle and decoder blocks exhibit increases in sparsity" is not well supported (xFVX):

The author provided a response in relation to change change in sparsity and sensitivity relates to data and learning, but did not directly answer the question regarding "the middle and decoder blocks exhibit increases in sparsity". This point was not addressed.

(5) unclear presentation, e.g., certain plots and analyses, are hard to follow (xFVX):

This point was not addressed.

(6) unclear what insight is offered by showing Euclidean distances correlating to distance between densities (xFVX):

The authors provided an explanation to this question. This is addressed.

(7) comparison with features from other models (MAE, DINO), particularly t-conditioned denoisers, and sensitivity to noise (xFVX):

The authors mentioned that it is possible for representations to be more sensitive to noise level with explicit t-conditioning by citing a related paper, but does not address the point regarding other representations.

(8) possible explanation of the "scene gist" due to architectural factors rather than denoising objective. (a9ev, GomQ):

Per the request of the reviewer, the authors added new quantitative results in Appendix which implies that the sparsity and selectivity emerge from trianing with denoising objective and not architectural design. Follow up of the discussion raised question regarding whether such behavior can arise from any objective outside of the denoising. Note that GomQ also asked a similar question regarding pretrained models. The authors did not respond to the follow-up question. While the a9ev did mention they would raise their score, the AC feels this is an important point to distinguish as it can change the conclusion of the paper.

(9) the work is not properly framed (G5Yd):

The authors provided a discussion to emphasize difference of conditional and unconditional models. This point is addressed.

(10) unclear connection between representation sparsity and denoising, claim appears to be severely under-justified, speculative, and mainly accepted by assumption (G5Yd):

The AC agree with the authors in that there exists connection between denoising and sparsity. This point has been addressed.

(11) unclear derivation and notation (G5Yd):

The authors clarified the notation in the comment. This point has been addressed.

(12) connection to probabilistic formulation of diffusion is ignored (G5Yd):

The authors explained that their work aims to examine properties of denoising models rather than on training or sampling. While there is literature on the topic, the AC recommends the authors to add such a discussion to make the paper more self-contained.

(13) unclear experimental set up and lack of rigorous experimental analysis, work is mainly empirical (G5Yd, GomQ):

The authors explain that their focus is on analyzing the internal representations of diffusion models and hence standard benchmarks are not directly applicable. The AC agrees with this point; however, the sentiment that the is lacking analysis has been shared across several reviewers. The AC advises the authors to consider more quantitative measurements to provide in-depth insights and analysis.

(14) conclusions are only drawn from one type of denoising architecture, specifically UNet (GomQ):

The authors mentioned that their work provides "mechanistic understanding of the most widely used architecture in diffusion and denoising research, and the mathematical principles we highlight apply broadly across architectures." To be able to make this claim, the AC agrees with the reviewer that they need to consider architectures beyond just a UNet.

(15) justification for normalized participation ratio as a measure of sparsity (GomQ):

The authors provide an explanation in the comment. This has been addressed.

(16) lack of rational for Algo. 1  (GomQ):

The authors provide an explantation in the comment. This has been addressed.

**Reviewer Scores:**

The AC has read the reviews and the rebuttal. Given the discussion thread, a9ev is expected to raise their score to a marginally above acceptance threshold while GomQ is expected to maintain rejection. As only some of the points were addressed for xFVX and GomQ, the AC expects them to also maintain their scores.

---

### Decision · Program_Chairs · 2026-01-26

Reject